# Dietary Betaine Interacts with Very Long Chain n-3 Polyunsaturated Fatty Acids to Influence Fat Metabolism and Circulating Single Carbon Status in the Cat

**DOI:** 10.3390/ani12202837

**Published:** 2022-10-19

**Authors:** Dennis E. Jewell, Matthew I. Jackson

**Affiliations:** 1Department of Grain Science and Industry, Kansas State University, Manhattan, KS 66506, USA; 2Hill’s Pet Nutrition, Topeka, KS 66617, USA

**Keywords:** cats, α-linolenic acid, EPA, DHA, betaine

## Abstract

**Simple Summary:**

The domestic cat can metabolize and thrive on a range of intakes of different dietary polyunsaturated fatty acids (PUFA). However, changes in the intake of PUFA have relatively unknown effects on concentrations of other fatty acids and metabolites. Similarly, the effect of increasing dietary betaine (which is a single carbon donor) on circulating concentrations of metabolites and fatty acids is relatively unreported. As might be expected, increasing intake of specific dietary fatty acids resulted in an increased concentration of that fatty acid and moieties containing that fatty acid. Dietary betaine increased concentration of many compounds associated with single carbon metabolism (e.g., dimethyl glycine, sarcosine, methionine) and many PUFA such as the n-6 PUFA linoleic acid (LA) and arachidonic acid (ARA) and the n-3 fatty acids α-linolenic acid (αLA), and docosahexaenoic acid (DHA). Dietary betaine interacted with the addition of dietary fish oil to dampen diet-induced increase of ARA while potentiating the increase of circulating DHA occurring with increased DHA dietary intake. Dietary betaine and fish oil also combined to reduce the circulating concentration of the renal toxin 3-indoxyl sulfate, suggesting a positive effect on the gut microbiota. These data suggest a positive effect of a daily betaine intake which exceeds 60 mg per kg body weight. The data also support an added benefit of a combined EPA+DHA daily intake of greater than 26 mg/kg body weight as well as a daily intake of 75 mg/kg body weight of alpha linolenic acid.

**Abstract:**

Six foods were used to evaluate the interaction of dietary betaine and n-3 PUFA in the cat. There was no ingredient added to the control food to specifically increase betaine or n-3 fatty acids. The experimental design was a 3 × 2 factorial (fatty acids were varied from the control food which had no added source of n-3 fatty acids, flax was included as a source of 18 carbon n-3, or menhaden fish oil as a source of EPA and DHA). Foods were then formulated using these three foods as a base with added betaine or without added betaine. Forty eight cats were used in this study. Equal numbers of cats were allotted by age and gender to each of the six dietary treatments. The cats were offered food amounts to maintain weight and consumed the food to which they were assigned for the length of the study (60 days). Metabolomics, selected circulating analytes and fatty acids were analyzed at the beginning and end of the feeding period. There was an increase in single carbon metabolites (betaine, dimethyl glycine, and methionine) with the consumption of dietary betaine. Betaine also increased the concentration of specific PUFA (ARA, αLA, DHA, and the sum of all circulating PUFA). The combination of dietary betaine and fish oil resulted in a reduction of circulating 3-indoxyl sulfate which suggests a renal benefit from their combined dietary presence.

## 1. Introduction

The placement of double bonds in relation to the methyl (omega) end and the number of double bonds together are important in the specific biological role of PUFA. In the cat, dietary increases in n-3 fatty acids (i.e., αLA, EPA and DHA), which have their methylene-interrupted series of double bonds located beginning at the 3rd carbon from the omega terminus are shown to influence immune response [1]. The overall effect of these fatty acids on the immune response has recently been reviewed [2]. Flax has been used as a source of soluble fiber and αLA for some time and is known to increase circulating αLA [3]. We have shown that increased dietary PUFA influenced circulating single carbon metabolites in the cat [4] and that there was an effect of dietary betaine on the response to dietary PUFA in the dog [5]; however, the interaction of increasing dietary PUFA and betaine is to our knowledge unreported in the cat.

Although αLA, EPA and DHA have been reported to suppress inflammation in cats the conversion of αLA through elongation and desaturation to EPA and DHA in the cat is somewhat controversial. It is clear that this capacity exists but EPA and DHA synthesis may be restricted to activity in the nervous tissue rather than the more common mammalian activity in the liver [6]. The n-6 family of fatty acids can similarly be elongated and desaturated so at least some LA can be converted to ARA [6]. The fatty acids LA, αLA, ARA, EPA and DHA are recognized as being necessary for inclusion in complete and balanced cat food [7]. It has been reported that adequate DHA status helps mitigate feline inflammation [8]. An increased consumption of EPA and DHA resulted in an improvement in the measurement of risk for calcium oxalate stone formation in cats [9]. Although there is no specific minimum required for overall concentrations of single carbon donors in cat food, there is a minimum balance of available single carbon donors needed such as folate, choline, methionine and betaine [10]. Although single carbon donors are known to be related to lipid metabolism and transport, particularly through roles in phosphatidylcholine production, it is not known the degree to which they interact with dietary lipids to alter their fasting circulating levels.

Betaine is both a single carbon methyl donor and an important osmolyte functioning both as an anti-inflammatory and anti-oxidant ingredient [11,12]. Betaine donates a methyl group to homocysteine to produce methionine and dimethyl glycine (through betaine-homocysteine methyltransferase). This allows methionine to act as a subsequent single carbon donor through S-adenosyl methionine (SAM) in a large number of reactions. The methylation of homocysteine to produce methionine resulting in an increase in SAM produces the antioxidant effect of betaine and does not require any specific antioxidant enzyme activity [13]. Betaine also moderates the immune response through the activity NF-κB and subsequent cytokine concentration regulation [14]. As an osmolyte betaine helps protect kidney health and function. This osmolyte role of betaine aids in the management of kidney disease through the reduction of hypertonicity [15]. We have previously published that incorporating betaine and fermentable fiber in foods designed to aid in the management of kidney disease resulted in a beneficial effect on body weight in cats [16] as well as reducing specific renal toxins including 3-indoxylsulfate [17]. We hypothesized that there was an interaction of betaine and PUFA when betaine status was changed following the addition of dietary PUFA in the cat. The effects of the interaction of dietary betaine and PUFA in the dog also supported the conclusion that this interaction was of scientific interest in the cat.

This study was designed to evaluate the null hypothesis that there would be no individual or interactive effects on the cat through changing the dietary status of PUFA and the single carbon donor betaine. The alternative hypothesis was that there would be an effect of both dietary betaine and PUFA on circulating fatty acids, specifically increasing the concentration of EPA and n-3 fatty acids. This was based on our original observation that dietary PUFA had influenced single carbon metabolism in the cat [4] and subsequently that dietary betaine and PUFA had an interaction together on circulating concentration of the n-3 PUFA and specifically on circulating EPA and total PUFA in then dog [5]. This design compared the effects of dietary betaine, flax, and fish oil alone and when present together. This factorial design makes possible the study of changing their individual dietary enhancements and their interactions. The response variables of selected health parameters, fatty acids, and metabolomics were chosen as indicators of the biological effects of changes in dietary PUFA, betaine and their interactions.

## 2. Materials and Methods

The study protocol was reviewed and approved by the Institutional Animal Care and Use Committee; Hill’s Pet Nutrition, Inc., Topeka, KS, USA (permit # CP833.0.0.0-A-F-MULTI-ADH-MUL-407-OTH) and complied with the guides for the care and use of laboratory animals from the National Institutes of Health, US National Research Council and US Public Health Service. All cats were cared for by animal care research technicians who were masked to the group identity during sample collection. Also, all sample analyses were completed by technicians who were masked to group identity of the cats associated with the sample. All cats were owned by the commercial funders. Forty eight cats (domestic short hair) were used in this study. During the 14 day washout period all cats were fed the control food (Table 1) which was a complete and balanced feline maintenance food. All foods had the same concentration of high protein sources which were chicken and corn gluten meal as well as the same concentrations of added vitamins. Cats were assigned to treatment by gender, weight and age. Each treatment had 4 neutered males and 4 spayed females. The average ages were: control group 5.9 years (range 1–12), control + flax group 5.7 (range 1–11), Control + Fishoil 5.8 (range 1–11), control + betaine 5.9 years (range 1–10) Control + betaine + flax 5.9 years (range 1–10), and control + betaine + fish oil 5.7 years (range 1–10). Each treatment was exclusively fed one food for the 60 days of the feeding study. Betaine was not added or at 0.5% where there was no specific PUFA source, or where flax (3% to increase αLA) or fish oil (to increase EPA and DHA) were the source of n-3 PUFA. The control food analyzed 736 mg/kg betaine as well as 0.15% αLA and had EPA and DHA below quantification (<0.02%). For the non-betaine supplemented foods, flax (αLA 0.52%) or fish oil (0.17% EPA, 0.11% DHA) were added. The three betaine supplemented foods were based on added betaine alone (6210 mg/kg), added betaine and added flax (αLa 0.64%) or with added betaine and fish oil (EPA 0.15%, DHA 0.11%). Choline (as choline chloride was added to all foods at 0.3% (dry matter basis). Carnitine (added as l-carnitine) was added to all foods at 560 mg/kg (dry matter basis). All foods were extruded. When betaine or flax was added it was in the dry mix before extrusion. Fish oil was added topically after extrusion. Food was available in dry form only. Ingredient use and analyzed nutrients are shown in Table 1. Each food was analyzed once using the following methods by a commercial laboratory (Eurofins Scientific, Inc., Des Moines, IA, USA): moisture—AOAC 930.15; protein—AOAC 2001.11; fat—AOAC 954.02; fiber—AOAC 962.09; and ash—AOAC 942.0. Mineral and fatty acid analyses were performed by the same commercial laboratory. Fatty acid (FA) concentrations were determined by gas chromatography of FA methyl esters. Food composition and ingredients are reported in Table 1. After a 16 h fast phlebotomy was completed. Blood samples were collected by venipuncture with approximately 5 mL of blood collected. Cats were given mild sedation by intramuscular injection of 2.9 mg/kg Tiletamine/zolazepam (Telazol) before blood samples were taken. Plasma samples were analyzed for metabolomics before the start of the trial foods and at the end of the study (60 days on test foods). Clinical blood chemistry was also completed at the same time points. A COBAS c501 module (Roche Diagnostics Corporation, Indianapolis, IN, USA) was used to complete the analysis the non-metabolomic blood values which were analyzed in serum. Metabolon (Morrisville, NC, USA) completed the analysis of the metabolomics as previously described [18,19]. A commercial laboratory (Eurofins Scientific, Inc., Des Moines, IA, USA) completed the dietary fatty acid analysis. The circulating fatty acid (FA) composition (non-esterified and bound fatty acid) was determined as previously described [20,21]. In brief serum concentrations of fatty acids were analyzed by use of gas chromatography of methylated circulating fatty acids. The extraction of fatty acids from the serum was in a mixture of chloroform and methanol. The chloroform was evaporated to dryness under a stream of nitrogen and the fatty acids were subsequently methylated. Water was added to extract fatty acid methyl esters. The methyl esters were separated and detected with a gas chromatograph and the subsequent peak areas used for quantification through comparison to standard curves generated using internal standards.

The ratio of the individual circulating fatty acids as compared to their daily intake was also calculated. This ratio was the result of the change in fatty acid concentration (µg/dL Final–µg/dL Initial) divided by the intake (g/day) of that fatty acid. This response variable has the advantage as compared to circulating concentration alone in that the variation associated with a change in food intake for each pet is taken into account (and thus fatty acid intake even though the same food is being consumed).

Physical evaluations and clinical indices were used to evaluate health before cats were assigned to the study. Four males and four females were assigned to one of six groups and then foods were randomly assigned to group. All cats were healthy and housed in group rooms (*n* = 8/room, each room was 4.27 m by 4.27 m) with access to glassed in porches (3.05 m by 2.13 m). Both the main room and the porch have natural lighting. Cats had daily opportunity for enrichment through interaction with animal care personnel and continual access to other cats, toys, climbing apparatus, and multiple levels. A controlled amount of food was offered daily in an amount calculated so that intake would maintain body weight. Each treatment consisted of a single room to keep cats from having access to a non-treatment food. The food amount offered to each cat was established from previous feeding studies. Food intake was monitored through electronic monitoring of each cat (each cat contains an identifying chip) so that access to food is controlled and measured individually although they are group housed.

The statistical program SAS 9.4 Proc MIXED (SAS Institute, Cary, NC, USA) was used for analysis. Split plot analysis was completed with gender as the whole plot. The complete factorial of PUFA source (non-added, flax or fish oil) and betaine (added or non-added) were used to define the sub-plot treatments. Significance was assigned if *p* ≤ 0.05 was observed. Because there were some differences discovered before treatments were assigned, the initial values were used as a covariant in evaluating the influence of treatments on the final fatty acid concentration. The post-hoc mean separation was completed using the pdiff statement and the Tukey-Kramer’s adjustment for multiple comparisons. The metabolomics analysis of significance required *p* < 0.05 and q < 0.10. Metabolomic analysis was performed on the log-transformed data with day zero being used for each cat as its own initial starting point. The reported changes over time are after conversion back from the log-transformed data. For the change over time or the difference between treatment groups at the end of the study to be significant a *p* value < 0.05 and a false discovery rate correction q value < 0.1 were used as cut offs. All of the foods met or exceed the requirements of AAFCO to be complete and balanced for adult cats for maintenance of health and body weight. They differed analytically through the addition of betaine and flax or fish oil as a source of PUFA. The three foods with no added betaine averaged 720 while the betaine supplemented foods averaged 6093 (mg/kg). The addition of flax increased αLA from an average of 0.16% to 0.58%. The addition of fish oil changed the EPA and DHA concentrations from below 0.02 to an average of 0.16% EPA and 0.11% DHA (Table 1).

## 3. Results

Body weight was not influenced by treatment either at the beginning or the end of the study. There was also no significant treatment influence on food intake or in final concentrations of circulating triglycerides, creatinine, urea, total protein or albumin (Table 2). There was an increased concentration of cholesterol in the cats fed foods containing betaine when compared to the unsupplemented controls with the final concentration increased in the betaine consuming cats as compared to the cats consuming non-betaine supplemented foods (*p* < 0.05). Each of the treatments of betaine consuming cats also had an increase (*p* < 0.01) in the final concentration of cholesterol as compared to the initial concentration of that treatment group.

PUFA source as well as dietary betaine influenced a number of circulating fatty acids (Table 3). Increasing flax and thus dietary αLA caused an increase in circulating αLA which was higher in the presence of dietary betaine than when added alone (*p* < 0.05). There was an increased concentration (*p* < 0.05) of DHA, DPA and EPA in response to dietary fish oil. The addition of dietary betaine increased the circulating concentration of DHA and ARA, circulating concentration of the sum of all n-6 and all PUFA, and total n-3 fatty acids as well as an increased ratio of EPA. Dietary betaine resulted in a decreased concentration of myristic acid and an overall increased concentration of stearic acid. There was an interaction of PUFA source and betaine such that in the presence of flax dietary betaine differentially increased the ratio of αLA while the presence of fish oil or betaine increaed the concentration and ratio of DHA. In the absence of dietary fish oil or flax dietary betaine increased the circulating concentration of oleic acid (Table 4).

There were 785 metabolites detected by the non-targeted metabolomics. The metabolites that were statistically selected which over time were changed in one or more treatments (*p* < 0.05, q < 0.1) are reported in Table 5, Table 6, Table 7, Table 8, Table 9, Table 10 and Table 11. There were 62 increased and 13 decreased in the cats fed the control food. There were 67 increased and 20 decreased in the cats fed the flax only supplemented food. There were 51 increased and 70 decreased in the cats fed the food supplemented with fish oil alone. There were 66 increased and 32 decreased in the cats which consumed the betaine alone supplemented food. There were 133 increased and 62 decreased in the cats consuming foods supplemented with betaine and flax. There were 113 increased and 114 decreased in the cats consuming betaine and fish oil.

The addition of betaine resulted in increased concentrations of all methylated glycines (sarcosine, dimethylglycine, betaine); however, glycine itself did not reach significance (Table 5). Further, betaine increased methionine and methionine metabolites including oxidation and catabolic products. There was a general time-dependent effect to increase cysteine and its related metabolites, as shown by increases of some members of this metabolite class in the CON group, but betaine did not consistently impact the metabolism of this sulfur amino acid. In contrast, inclusion of flax seed in the foods increased cysteine metabolites regardless of the presence of betaine. Particularly, flax inclusion increased homocysteine to levels several fold higher than other treatment groups, and was the only treatment to increase cystathionine or lanthionine. Feline species generate a unique sulfur-containing peptide termed felinine that is derived from cysteine via the intermediacy of glutathione. While glutathione was not impacted by treatments (data not shown), fish oil in the absence of betaine was the only treatment to increase all felinine-related metabolites; co-inclusion of betaine abrogated the effect of fish oil on felinine metabolism.

Diacylglycerides (DAG) are intermediate products of lipolysis, typically produced by the action of phospholipase C on phospholipids or that of lipoprotein lipase on triglycerides. Betaine inclusion alone increased DAG, and this effect was increased with flax co-inclusion (Table 6). In contrast, fish oil in the absence of betaine increased DAG, but co-inclusion of betaine with fish oil largely eliminated increased DAG. Dicarboxylates (DCB) are products of fatty acid oxidation by peroxisomes. The addition of betaine decreased several DCB and this effect was maintained in the presence of fish oil co-inclusion, but inclusion of flax alongside betaine appeared to decrease the capacity of betaine to impact DCB.

Acylcarnitines (ACN) are fatty acid conjugates of carnitine, a moiety which facilitates the transport of fatty acids into mitochondria for oxidative phosphorylation. Fish oil alone had minimal impact on ACN, except it increased those ACN which contain fatty acids (or their metabolites) found in fish oil (Table 7). In contrast, flax inclusion alone increased several ACN of the intermediate, long and very long chain classes. Surprisingly, betaine co-inclusion alongside these two fat sources decreased ACN relative to the condition without betaine such that the increase in ACN by flax was abrogated and the fish oil + betaine condition manifested decreased ACN. Despite this effect of betaine to influence the effect of flax or fish oil on ACN, betaine alone did not have an effect on ACN.

Choline-containing phospholipid levels were influenced to a greater degree by fish oil alone than they were by flax inclusion alone (Table 8). As expected, the phospholipids which contained flax or fish oil-derived fatty acids were increased in the treatment groups which received these fat sources in the diet. Thus, αLA-containing phospholipids were increased in the two groups receiving flax and DHA-containing phospholipids were increased in the two groups receiving fish oil. Choline is a quaternary amine that can be converted to betaine by oxidation, and thus betaine and choline share many biological functions. Although choline, choline phosphate and glycerophosphorylcholine were not increased by betaine, inclusion of betaine alone increased a broad swath of choline-containing phospholipids. Betaine’s tendency to increase choline-containing phospholipid levels was also manifest in the flax + betaine treatment condition, but inclusion of fish oil alongside betaine largely reversed the effect of betaine. Similar effects were observed for ethanolamine-containing phospholipids (Table 9): Fish oil alone impacted this class of lipids to a greater extent than did flax alone, betaine tended to increase ethanolamine-containing phospholipids (although to a lesser extent than it did for the choline-containing phospholipids), and flax preserved the effect of betaine while fish oil abrogated or reversed the effect of betaine. While choline phospholipids can be synthesized from ethanolamine phospholipids, and betaine can be synthesized from choline, inositol is another hydrophilic moiety present in some phospholipids that is in a chemical class which is distinct from the aforementioned molecules. Intriguingly, while betaine alone did not affect the levels of any inositol-containing phospholipids (Table 9), the co-inclusion of fat source alongside betaine resulted in an increase (flax) or mixed increase and decrease (fish oil) of these lipids.

Sphingolipids are a special class of phospholipids, most of whose members contain choline as the hydrophilic moiety. There was a time-dependent increase in sphingolipid precursors and derivatives including sphingadienine, sphinganine and sphingosine, as well as phosphorylated congeners of the latter two molecules, as evidenced by higher levels in the control group (Table 10). However, both flax and fish oil prevented this increase over time. The groups receiving foods with betaine and sources of n-3 fatty acids also showed an apparent increase over the course of the study. Betaine alone increased sphingomyelins. While flax alone did not manifest an increase in sphingomyelins, the combination of flax with betaine increased the levels of more than two dozen of these structural lipids. Fish oil alone impacted sphingomyelin levels; while most of the changes were increases, fish oil also led to the greatest number of decreased sphingomyelins. The combination of fish oil and betaine abrogated some of the decreases in sphingomyelins observed with fish oil alone and raised the number of sphingomyelins whose levels were increased from baseline. Thus, there appeared to be an interaction between betaine and n3 PUFA sources to increase circulating sphingomyelins.

Examination of the levels of classes of endocannabinoids and N-acylated amino acids which were significantly impacted by diets revealed differences in the impacts of flax versus fish oil, and an apparent interactive effect of betaine (Table 11). There were three ethanolamide class endocannabinoids that reached criteria for significance; however, no 2-acylglycerol type endocannabinoids reached significance in any group. Oleoyl ethanolamide appeared to manifest a time-dependent increase, as evidenced by higher levels from baseline in the control group. Flax supplementation by itself further increased pamitoyl, although not stearoyl ethanolamide, but fish oil alone was without effect on any of the observed ethanolamides. Betaine itself was largely without effect, only increasing one N-acylated amino acid (N-stearoylserine). However, when combined with flax it resulted in an increase of the sole ethanolamide which flax alone had not changed, while maintaining the increase in the two ethanolamides which flax alone had increased. Altogether flax combined with betaine increased three ethanolamides from baseline and decreased two N-acylated amino acids. In contrast, while fish oil alone was completely without effect on any ethanolamdes or N-acylated amino acids, the combination of fish oil and betaine robustly decreased several N-acylated amino acids (while increasing only one: N-stearoylserine). Further, fish oil eliminated the time-dependent increase in oleoyl ethanolamide observed in the control group, and the combination of fish oil and betaine resulted in a decrease in oleoyl ethanolamide. From these data, there was an apparent difference in the impact of betaine on the effects of the two different n-3 PUFA sources.

Table 12 provides the significant effects of treatments on circulating bile acids and cholesterol as well as metabolites in the cholesterol synthesis pathway. The most prominent effect was that of fish oil (alone or in combination with betaine) to decrease bile acids. Intriguingly, betaine alone increased cholesterol, an effect which was also manifest in the groups which received betaine in combination with either flax or fish oil. Although fish oil combined with betaine decreased levels of both cholesterol precursors, the levels of total cholesterol itself were still increased.

## 4. Discussion

Dietary permethylated lipotropes (choline, methionine, betaine) are so named for their effect to mobilize hepatic fat depots, a process of hepatic export of triglycerides in a cholesterol-dependent manner [22]. Although few of the non-fatty acid metabolites changed with treatment, there was an overall effect of dietary betaine in increasing circulating concentrations of total cholesterol, which may be in concert with its capacity as a lipotrope. This increase (16%) did not move the cats out of the normal reference range for cholesterol (100–277 mg/dL). Dietary betaine has been shown to increase circulating cholesterol concentrations in pigs [23] and in dogs [5]. A recent meta-analysis concluded that in humans a dietary betaine intake of 4 g/day “moderately increases total cholesterol levels” [24] which had previously been reported [25]. However, when betaine was consumed as part of a whole grain there was an increase in circulating betaine concentration which was associated with a decrease in plasma total and LDL cholesterol [26]. In rats, the mechanism of action for dietary betaine which resulted in increasing circulating cholesterol was through a promotion of de novo synthesis and bile acid conversion [27]. These data are consistent with such a mechanism existing in cats.

In terms of specific increases in fatty acids which had been purposely enriched in the intervention diets, there is an expected increase in circulating concentrations of dietary EPA and DHA with the inclusion of dietary fish oil [28]. This has also specifically been shown in cats [3,6]. The increased concentration of circulating EPA and DHA results in a changed cell membrane composition of fatty acids and a reduced inflammatory profile [29,30]. This study shows that increasing the dietary fatty acids of αLA, EPA or DHA results in increased circulating concentrations of those fatty acids while adding betaine increased the concentrations of a significant number of PUFA (LA, αLA, ARA, DHA, sum of n-6 and sum of PUFA) clearly showing that single carbon metabolism influences fatty acid metabolism in the cat. This is similar to that which was shown in quail where dietary betaine increased circulating EPA [31] and in the dog where dietary betaine resulted in increased circulation of EPA, DPA and sum of n-3 fatty acids [5]. The different species response suggests a unique control of these circulating fatty acids. In this study there was an increased concentration of αLA in the cats fed dietary betaine and flax as compared to cats eating all other foods. Because αLA modulates immune response in rats [32], humans [33,34] and swine [35], as well as cats [1] it is likely that betaine and flax would work through αLA to reduce inflammation. However, dietary betaine alone, especially in the absence of added PUFA resulted in an increased circulating concentration of ARA and total n-6 fatty acids which may result in a pro-inflammatory cytokine profile. The increased concentration of EPA and DHA in response to dietary fish oil would be expected to reduce feline inflammation [1]. The changes in circulating fatty acids as influenced by betaine show a somewhat different response in the cat than those reported in the dog [5] in that in the cat there was no response to added αLA in circulating EPA (which was moved to be intermediate between control and added fish oil in the dog). Although betaine influenced circulating PUFA in both species the effect of αLA was more significant in the dog. This study cannot differentiate causes for these circulating changes but it suggests the value of further research establishing where (or if) betaine influences desaturation and elongation of fatty acids.

The cat as a species is particularly reliant on single carbon metabolism [10]. The current report shows how that metabolism is influenced by betaine; as previously shown in the dog [5,11], these data show that dietary betaine increased circulating concentrations of methionine, dimethyl glycine and sarcosine. The addition of betaine as a single carbon source had therefore the expected influence on metabolites which are directly related to betaine through single carbon metabolism (e.g., increases in dimethyl glycine, sarcosine, and methionine). Particularly, the reduction in felinine and increase in methionine by betaine consumption indicates that there was shunting of sulfur towards remethylation by betaine:homocysteine methyltransferase rather than cysteine (and subsequent felinine) production. Interestingly, as betaine is the methyl source which combines with homocysteine to produce methionine (and dimethyl glycine) there was an unexpected increase in homocysteine in all treatments except in the cats consuming betaine + fish oil. This dietary change seems to have also influenced the transsulfuration pathway of homocysteine degradation. This pathway is the condensing of homocysteine and serine to form cystathionine which is split into alpha-ketobutyrate and cysteine [36]. A possible reason for the increased homocysteine in the cats consuming foods containg flax is that flax is increasing cystathionine through inhibition of the enzyme cystathionase as was shown in chickens [37]. Similar cystathionine results were reported in rats fed a flax anti-pyridoxine factor resulting in both an increased cystathionine concentration and an increase in homocysteine [38]. The reduction of alpha-ketobutyrate in the cats fed betaine and fish oil could be a positive aid in the management of feline urolithiasis in that this metabolite has been found in this colony of cats to be increased in cats that formed calcium oxalate uroliths (data not shown).

The reduced concentrations of the dicarboxylate compounds in the presence of dietary betaine is similar to the reduction associated with dicarboxylate compounds when cats were fed a high dietary protein as compared to a high fat food [39]. The shift in the dicarboxylate compounds with the addition of dietary betaine as compared to dietary fat or protein suggests that betaine metabolism may be acting on energy metabolism to enhance fatty acid breakdown and off set amino acid metabolism. This energy metabolism shift may be part of the explanation for a betaine containing food enhancing total body weight in renal insufficient cats [40]. Further evidencing the impact of betaine on fat metabolism, the addition of dietary betaine, especially in the presence of dietary fish oil, resulted in a reduction of a significant number of acylcarnitines which is similar to that seen in the dog [5]. This reduction was also observed in rats responding to a high-fat food and dietary fat consumption. In the rat it was concluded that there was a shift between circulating and organ-bound carnitine [41]. This shift could be a part of the metabolism energy partitioning which resulted in the reduced concentration of dicarboxylate compounds observed as the increased organ bound carnitine moieties would participate in fatty acid oxidation and a general shift to fatty acid utilization.

There is an increased DAG concentration resulting from betaine alone or in the cats consuming the combined betaine and flax food. This suggests an increased lipolysis. However, there was no corresponding shifts in food intake or body weight so energy mobilization if present was not influencing these foundational outcomes. Further research is needed to further evaluate the role of betaine in shifting energy metabolism.

In contrast to the effect of betaine to reduce circulating fatty acid catabolic intermediates, betaine largely led to an increase in circulating levels of structural lipids. Of the choline containing phospholipid compounds that were influenced by dietary treatment there was an increase in the cats consuming control food plus betaine and those consuming control + betaine + flax (31 increased, 3 decreased) as compared to the cats consuming betaine + fish oil (12 increased, 15 decreased). It is well known that the dietary compounds betaine and choline, which both contain labile methyl groups, mutually enhance the circulating concentration of each other [10]. Therefore, it is not unexpected that the addition of dietary betaine enhances choline containing compounds. The offsetting effects of betaine and fish oil might be explained by betaine enhancing the choline component while the effect of dietary fish oil is to reduce the concentrations of linoleic, αLA, palmitoleic as was observed in the cats eating the dietary fish oil in this study. This effect of dietary fish oil on specific fatty acid concentrations could explain the effect of the reduction of the ethanolamine containing compounds (as well as the reasonable increases in the DHA containing compounds). These changes highlight the need for research investigating the response in cats to changes in betaine and choline independently and together. 

The sphingolipids that changed were generally increased in the cats consuming dietary betaine (95 increased, 6 decreased). This is reasonably the result of increased concentrations of sphinganine and sphingosine. This may be a benefit as it has been suggested that these play a role in host defense of microbial attack [42]. This positive immune response could be one of the supporting causes of dietary betaine having a positive immune response in the dog [43]. The reduction of 7-HOCA in the cats consuming food enhanced with dietary betaine and fish oil may be of benefit in reducing the inflammation associated with aging and cell senescence in a cell culture model [44]. This reduction was also associated with a benefit in controlling the risk of Alzheimer’s disease in humans [45]. This study suggests an enhancement through control of inflammation. However, more work is needed to investigate the interaction of betaine with the less inflammatory fatty acids to understand this response.

Regarding dietary betaine and PUFA-based partitioning of fats into signaling lipids, there was minimal impact of betaine, fish oil or flax alone but there was an interaction between betaine and fish oil to nearly uniformly decrease circulating endocannabinoids and N-acylated amino acids. This interaction was not present when betaine was supplemented alongside flax seed. Thus, it would appear that the impact of betaine in conjunction with PUFA to impact circulating levels of these signaling lipids is dependent upon either chain length, degree of unsaturation or both. We had previously observed that fish oil had no large effect on circulating endocannabinoids and N-acylated amino acids in cats [18], and this current study affirms those findings. However, also from that previous study [18] we reported that the combination of medium chain triglycerides and fish oil broadly decreased circulating endocannabinoids and N-acylated amino acids. The current findings with betaine expand upon the nutritional scenarios under which fish oil is able to synergistically decrease circulating endocannabinoids and N-acylated amino acids. We’ve reported that gut-microbiome active dietary interventions (e.g., resistant starch) can also modulate circulating status of endocannabinoids and N-acylated amino acids in cats [19]. Since these signaling lipids play important roles in feline physiology [46,47,48,49,50] and increasing evidence indicates they are subject to nutritional modulation [48,49], the current report will aid in determining feline dietary interventions which are appropriate for lifestage or health status that can be benefitted by changes in endocannabinoid and N-acylated amino acid status.

## 5. Conclusions

These data show that in the cat betaine supplementation appears to influence fat metabolism and partitioning by differentially modulating levels of total cholesterol, fatty acid catabolic intermediates, structural lipids and signaling lipids. Some of these specific metabolomic changes suggest dietary betaine and fatty acid source influence indicators of health in cats. The composite effect of dietary betaine and fish oil suggests a benefit for reduction of feline inflammation and possible enhanced immune response, perhaps through modulation of endocannabinoid tone.

## Figures and Tables

**Table 1 animals-12-02837-t001:** Food composition and ingredient mix of pre-trial and test foods (grams/100 grams unless otherwise stated).

Analyte or Ingredient	Control (and Pretrial)	Control + Flax	Control + Fish Oil	Control + Betaine	Control + Betaine + Flax	Control + Betaine + Fish Oil
Corn Gluten Meal	32.45	32.45	32.45	32.45	31.95	32.45
Wheat	32.45	29.42	32.45	31.95	29.42	31.95
Dried Chicken	9.65	9.65	9.65	9.65	9.65	9.65
Pork Fat	15.23	15.23	14.23	15.23	15.23	14.23
Beet Pulp	2.5	2.5	2.5	2.5	2.5	2.5
Chicken Liver Digest	2.5	2.5	2.5	2.5	2.5	2.5
Flax Seed	0	3.03	0	0	3.03	0
Fish Oil	0	0	1	0	0	1
Betaine				0.5	0.5	0.5
L-Carnitine (10%) ^φ^	0.45	0.45	0.45	0.45	0.45	0.45
Choline Chloride ^χ^ (70%)	0.42	0.42	0.42	0.42	0.42	0.42
Taurine	0.18	0.18	0.18	0.18	0.18	0.18
dL Methionine	0.05	0.05	0.05	0.05	0.05	0.05
Lysine	0.04	0.04	0.04	0.04	0.04	0.04
Vitamins, Minerals, and processing aids	4.08	4.08	4.08	4.08	4.08	4.08
Moisture	6.78	6.30	5.86	7.07	6.12	6.97
Protein	33.38	34.50	33.50	33.63	31.88	32.88
Fat	19.22	21.62	20.72	19.64	19.94	19.20
Atwater Energy ^€^ (kcal/kg)	3997	4127	4100	4009	4051	3990
Ash	4.87	4.87	4.81	4.75	4.93	4.77
Crude Fiber	1.6	1.8	1.8	1.7	1.7	1.7
Calcium	0.81	0.81	0.80	0.82	0.83	0.81
Phosphorus	0.69	0.70	0.68	0.68	0.68	0.69
Sodium	0.30	0.32	0.30	0.30	0.30	0.31
Betaine (mg/kg)	736	727	696	6210	6440	5630
Capric acid [10:0]	<0.02	<0.02	<0.02	<0.02	<0.02	<0.02
Lauric acid [12:0]	<0.02	<0.02	0.02	<0.02	<0.02	<0.02
Myristic acid [14:0]	0.20	0.22	0.26	0.22	0.21	0.24
Palmitic acid [16:0]	4.11	4.54	4.12	4.40	4.35	3.89
Palmitoleic acid [16:1]	0.50	0.55	0.57	0.53	0.53	0.53
Stearic acid [18:0]	1.98	2.23	1.94	2.15	2.13	1.82
Oleic acid [18:1]	6.81	7.62	6.70	7.30	7.35	6.29
Arachidic acid [20:0]	0.04	0.04	0.04	0.04	0.04	0.04
LA [18:2 (n-6)]	3.50	3.81	3.47	3.63	3.73	3.27
αLA [18:3 (n-3)]	0.15	0.52	0.17	0.16	0.64	0.17
ARA [20:4 (n-6)]	0.08	0.08	0.09	0.08	0.08	0.09
EPA [20:5 (n-3)]	<0.02	<0.02	0.17	<0.02	<0.02	0.15
DPA [22:5 (n-3)]	<0.02	<0.02	0.04	<0.02	<0.02	0.03
DHA [22:6 (n-3)]	<0.02	<0.02	0.11	<0.02	<0.02	0.11
SFA ^£^	6.45	7.18	6.51	6.94	6.86	6.12
MUFA ^¥^	7.51	8.39	7.50	8.06	8.09	7.02
PUFA ^π^	3.97	4.69	4.38	4.14	4.71	4.12
(n-6) FA ^Ω^	3.74	4.07	3.73	3.89	3.98	3.51
(n-3) FA ^θ^	0.19	0.57	0.53	0.20	0.69	0.51
(n-6):(n-3) ratio	19.7	7.1	7.0	19.5	5.8	6.9

^χ^ Purchased from Balchem Corporation New Hampton, NY, USA; ^φ^ Purchased from Lonza, Basel, Switzerland; ^€^ Calculated from analyticals using modified Atwater numbers (kcal/g of 3.5 for protein, 8.5 for fat and 3.5 for nitrogen free extract); ^£^ Sum of the saturated fatty acids: 8:0 + 10:0 + 11:0 + 12:0 + 14:0 + 15:0 + 16:0 + 17:0 + 18:0 + 20:0 + 22:0 + 24:0; ^¥^ Sum of the monounsaturated fatty acids: 14:1 + 15:1 + 16:1 + 17:1 + 18:1 + 20:1 + 22:1 + 24:1; ^π^ Sum of the polyunsaturated fatty acids: 18:2 (n-6) + 18:3 (n-6) + 18:3 (n-3) + 18:4 (n-3) + 20:2 (n-6) + 20:3 (n-6) + 20:3 (n-3) + 20:4 (n-6) + 20:4 (n-3) + 20:5 (n-3) + 21:5 (n-3) + 22:2 (n-6) + 22:4 (n-6) + 22:5 (n-6) + 22:5 (n-3) + 22:6 (n-3); ^Ω^ Sum of the (n-6) fatty acids; ^θ^ Sum of the (n-3) fatty acids.

**Table 2 animals-12-02837-t002:** Body weight and selected serum biochemistries from serum biochemical profiles (values are lsmeans ± standard errors).

Analyte	Control	Control+ Flax	Control + Fish Oil	Control + Betaine	Control + Flax + Betaine	Control + Fish Oil + Betaine
Body Weight (kg) Initial	4.94 ± 0.53	5.36 ± 0.53	5.37 ± 0.53	5.00 ± 0.53	5.73 ± 0.54	5.41 ± 0.53
Body Weight (kg) Final	5.25 ± 0.58	5.72 ± 0.58	5.56 ± 0.58	5.20 ± 0.58	6.10 ± 0.59	5.44 ± 0.58
Food intake (g/day)	57.3 ± 9.4	65.1 ± 9.4	76.2 ± 9.4	54.1 ± 9.4	68.3 ± 9.7	55.5 ± 9.4
Albumin (mg/dL) Initial	3.52 ± 013	3.59 ± 0.13	3.60 ± 0.13	3.66 ± 0.13	3.70 ± 0.13	3.59 ± 0.13
Albumin (mg/dL) Final	3.29 ± 011	3.43 ± 0.11	3.30 ± 0.11	3.33 ± 0.11	3.40 ± 0.12	3.29 ± 0.11
Total Protein (mg/dL) Initial	6.84 ± 0.15	6.95 ± 0.15	6.81 ± 0.15	6.66 ± 0.15	6.74 ± 0.16	6.98 ± 0.15
Total Protein (mg/dL) Final	6.92 ± 0.17	6.99 ± 0.17	6.86 ± 0.17	6.44 ± 0.17	6.55 ± 0.19	6.74 ± 0.17
Urea Nitrogen (mg/dL) Initial	23.5 ± 1.1	21.0 ± 1.1	18.9 ± 1.1	22.0 ± 1.1	22.0 ± 1.2	22.6 ± 1.1
Urea Nitrogen (mg/dL) Final	25.8 ± 1.3	24.9 ± 1.3	23.0 ± 1.3	23.1 ± 1.3	22.8 ± 1.4	20.8 ± 1.3
Creatinine (mg/dL) Initial	1.14 ± 0.07	1.09 ± 0.07	1.17 ± 0.07	1.11 ± 0.07	1.18 ± 0.07	1.18 ± 0.07
Creatinine (mg/dL) Final	1.13 ± 0.07	1.14 ± 0.07	1.16 ± 0.07	1.13 ± 0.07	1.19 ± 0.07	1.20 ± 0.07
Triglycerides (mg/dL) Initial	34.3 ± 5.3	43.9 ± 5.3	28.9 ± 5.3	32.0 ± 5.3	39.4 ± 5.7	27.4 ± 5.3
Triglycerides (mg/dL) Final	44.5 ± 6.4	45.5 ± 6.4	33.7 ± 6.4	38.1 ± 6.4	46.1 ± 6.8	27.4 ± 6.4
Cholesterol (mg/dL) ^¥^ Initial	127.4 ± 13.6	140.5 ± 13.6	151.1 ± 13.6	183.0 ± 13.6	179.9 ± 14.4	155.0 ± 13.6
Cholesterol (mg/dL) ^¥^ Final	127.3 ± 14.9	140.6 ± 14.9	165.3 ± 14.9	210.0 ± 14.9	208.6 ± 15.9	184.4 ± 14.9

^¥^ main effect of betaine *p* < 0.01. Cats consuming betaine showed a higher concentration and had a higher (*p* < 0.01) change (final-initial) than cats not consuming added betaine.

**Table 3 animals-12-02837-t003:** Circulating concentration (mg/dL) of initial polyunsaturated fatty Acids, delta concentration over the course of the study (Final–Initial), or ratio of change in circulating fatty acid concentration divided by daily intake (µg/dL/mg daily intake). Values are lsmeans ± standard errors.

Analyte	Control	Control + Flax	Control + Fish Oil	Control + Betaine	Control + Flax + Betaine	Control + Fish Oil + Betaine	F-Test *p* Value ^Ω^
LA [18:2 (n-6)] Initial	40.8 ± 4.1	49.3 ± 4.1	52.5 ± 4.1	52.6 ± 3.8	61.1 ± 4.1	47.6 ± 4.0	0.07
LA [18:2 (n-6)] Delta	−3.1 ± 2.7 ^b^	−2.0 ± 2.6 ^b^	−8.5 ± 2.6 ^b^	13.0 ± 2.6 ^a^	10.0 ± 2.9 ^a^	−5.5 ± 2.6 ^b^	<0.01 P, B
αLA [18:3 (n-3)] Initial	1.4 ± 0.2 ^b^	1.8 ± 0.2 ^a,b^	1.8 ± 0.2 ^a,b^	2.1 ± 0.1 ^a^	2.2 ± 0.2 ^a^	2.0 ± 0.2 ^a,b^	<0.01 B
αLA [18:3 (n-3)] Delta	−0.46 ± 0.11 ^c^	0.18 ± 0.11 ^b^	−0.61 ± 0.10 ^c^	−0.02 ± 0.11 ^b^	0.75 ± 0.12 ^a^	−0.63 ± 0.11 ^c^	<0.01 P, B, P*B
αLA ratio	−44.1 ± 11.1 ^c^	−0.3 ± 10.3 ^a,b^	−51.8 ± 10.3 ^c^	−13.5 ± 10.6 ^b,c^	32.7 ± 11.3 ^a^	−66.6 ± 10.3 ^c^	<0.01 P, P*B
ARA [20:4 (n-6)] Initial	15.2 ± 1.4	17.0 ± 1.4	16.5 ± 1.4	19.2 ± 1.4	20.5 ± 1.5	19.0 ± 1.4	<0.01 B
ARA [20:4 (n-6)] Delta	0.54 ± 0.67 ^c^	0.81 ± 0.64 ^c^	0.61 ± 0.65 ^b,c^	4.18 ± 0.65 ^a^	3.63 ± 0.71 ^a,b^	1.10 ± 0.64 ^b,c^	<0.01 P, B, P*B
EPA [20:5 (n-3)] Initial	0.3 ± 0.1	0.3 ± 0.1	0.3 ± 0.1	0.2 ± 0.1	0.3 ± 0.1	0.3 ± 0.1	0.16
EPA [20:5 (n-3)] Delta	−0.06 ± 0.22 ^b^	−0.04 ± 0.22 ^b^	5.14 ± 0.22 ^a^	−0.03 ± 0.22	−0.05 ± 0.24 ^b^	4.53 ± 0.22 ^a^	<0.01 P
EPA ratio	−7.7 ± 3.7 ^b^	−1.8 ± 3.7 ^b^	44.5 ± 3.8 ^a^	−5.0 ± 3.7 ^b^	−3.6 ± 4.0 ^b^	48.4 ± 3.8 ^a^	<0.01 P
DPA [22:5 (n-3)] Initial	1.1 ± 0.1	1.1 ± 0.1	1.0 ± 0.1	1.2 ± 0.1	1.2 ± 0.1	1.1 ± 0.1	0.43
DPA [22:5 (n-3)] Delta	−0.05 ± 0.9 ^c^	−0.07 ± 0.06 ^c^	0.96 ± 0.06 ^a^	0.08 ± 0.07 ^c^	0.06 ± 0.07 ^c^	0.72 ± 0.06 ^b^	<0.01 P, P*B
DHA [22:6 (n-3)] Initial	2.3 ± 0.3	2.1 ± 0.3	2.0 ± 0.3	2.4 ± 0.3	2.6 ± 0.3	2.0 ± 0.3	0.77
DHA [22:6 (n-3)] Delta	−0.37 ± 0.29 ^b^	−0.42 ± 0.29 ^b^	5.45 ± 0.29 ^a^	−0.12 ± 0.29 ^b^	−0.10 ± 0.32 ^b^	6.32 ± 0.29 ^a^	<0.01 P, B
DHA ratio	−71.3 ± 21.3 ^c^	−61.7 ± 21.3 ^b,c^	60.4 ± 21.4 ^b^	−8.6 ± 21.4 ^b,c^	−10.7 ± 23.0 ^b,c^	98.0 ± 21.4 ^a^	<0.01 P, B
Sum of n-3 ^£^ Initial	4.99 ± 0.46	5.21 ± 0.46	5.04 ± 0.46	5.94 ± 0.44	6.27 ± 0.50	5.22 ± 0.46	0.31
Sum of n-3 ^£^ Delta	−0.67 ± 0.44 ^b^	−0.26 ± 0.44 ^b^	11.03 ± 0.44 ^a^	−0.30 ± 0.48 ^b^	0.41 ± 0.48 ^b^	10.91 ± 0.44 ^a^	<0.01 P
Sum of n-6 ^θ^ Initial	62.7 ± 5.6 ^b^	73.5 ± 5.6 ^a,b^	75.5 ± 5.6 ^a,b^	79.8 ± 5.3 ^a,b^	90.1 ± 5.9 ^a^	73.7 ± 5.6 ^a,b^	0.02 B
Sum of n-6 ^θ^ Delta	1.89 ± 3.83 ^a,b^	−0.69 ± 3.83 ^b^	2.37 ± 3.83 ^a,b^	16.76 ± 3.83 ^a^	8.80 ± 4.09 ^a,b^	7.24 ± 3.83 ^a,b^	<0.01 B
Sum of PUFA ^¥^ Initial	67.7 ± 5.9 ^b^	78.7 ± 5.9 ^a,b^	80.5 ± 5.9 ^a,b^	86.5 ± 5.6 ^a,b^	96.3 ± 6.2 ^a^	78.9 ± 5.9 ^a,b^	0.02 B
Sum of PUFA ^¥^ Delta	1.23 ± 4.00 ^b^	−0.96 ± 4.00 ^b^	13.40 ± 4.00 ^a,b^	16.45 ± 4.00 ^a,b^	9.22 ± 4.28 ^a,b^	18.2 ± 4.00 ^a^	<0.01 P, B

^a,b,c^ Means with different superscripts are different (*p* ≤ 0.05); ^£^ Sum of n-3 fatty acids defined below; ^θ^ Sum of n-6 fatty acids defined below; ^¥^ Sum of the polyunsaturated fatty acids: 18:2 (n-6) + 18:3 (n-6) + 18:3 (n-3) + 20:2 (n-6) + 20:3 (n-6) + 20:3 (n-3) + 20:4 (n-6) + 20:4 (n-3) + 20:5 (n-3) + 21:5 (n-3) + 22:5 (n-3) + 22:6 (n-3); ^Ω^ The *p* value is for the f test of the model, P is a main effect of PUFA source; B is a main effect of betaine; P*B is a significant effect of the interaction of betaine and polyunsaturated fat source; P and B appear when the f test showed a significance *p* ≤ 0.05.

**Table 4 animals-12-02837-t004:** Circulating concentration (mg/dL) of saturated and monounsaturated fatty acids profiles (values are lsmeans ± standard errors).

Analyte	Control	Control + Flax	Control + Fish Oil	Control + Betaine	Control + Flax + Betaine	Control + Fish Oil + Betaine	F-Test *p* Value ^Ω^
Myristic acid [14:0] Initial	0.33 ± 0.03	0.37 ± 0.04	0.40 ± 0.05	0.46 ± 0.05	0.53 ± 0.05	0.42 ± 0.05	0.02 B
Myristic acid [14:0] Delta	−0.07 ± 0.03	−0.05 ± 0.03	−0.01 ± 0.03	−0.04 ± 0.03	−0.16 ± 0.04	−0.11 ± 0.03	0.04 B
Palmitic acid [16:0] Initial	21.2 ± 1.7 ^b^	23.7 ± 1.7 ^a,b^	25.6 ± 1.7 ^a,b^	27.6 ± 1.7 ^a,b^	30.2 ± 1.8 ^a^	24.0 ± 1.7 ^a,b^	0.01
Palmitic acid [16:0] Delta	−0.39 ± 1.26 ^a,b^	−1.35 ± 1.20 ^b^	1.15 + 1.18 ^a,b^	4.86 ± 1.21 ^a^	1.55 ± 1.36 ^a,b^	0.91 ± 1.19 ^a,b^	0.02 B
Palmitoleic acid [16:1] Initial	1.1 ± 0.1 ^b^	1.3 ± 0.1 ^b,c^	1.2 ± 0.1 ^a,b^	1.4 ± 0.1 ^a,b^	1.6 ± 0.1 ^a^	1.2 ± 0.1 ^a,b^	0.02
Palmitoleic acid [16:1] Delta	−0.04 ± 0.10 ^a,b^	−0.01 ± 0.10 ^a,b^	−0.19 ± 0.10 ^b^	0.25 ± 0.10 ^a^	−0.05 ± 0.12 ^a,b^	−0.28 ± 0.10 ^a,b^	<0.01 P
Stearic acid [18:0] Initial	39.1 ± 3.2	44.8 ± 3.2	44.7 ± 3.2	46.8 ± 3.2	53.3 ± 3.4	45.7 ± 3.2	0.03 B
Stearic acid [18:0] Delta	0.67 ± 2.06	1.07 ± 1.96	3.24 ± 1.96	8.60 ± 1.96	9.32 ± 2.22	5.60 ± 1.96	<0.01 B
Oleic acid [18:1] Initial	20.2 ± 1.7 ^b^	22.9 ± 1.7 ^a,b^	23.4 ± 1.7 ^a,b^	26.0 ± 1.7 ^a,b^	30.1 ± 1.8 ^a^	22.0 ± 1.7 ^a,b^	<0.01 B, P*B
Oleic acid [18:1] Delta	−0.23 ± 1.28 ^b^	−0.44 ± 1.21 ^b^	−1.58 ± 1.21 ^b^	6.88 ± 1.22 ^a^	3.76 ± 1.46 ^a,b^	−2.67 ± 1.23 ^b^	<0.01 P, B, P*B

^Ω^ The p value is for the f test of the model; ^a,b,c^ Means that do not share a superscript are different *p* < 0.05; P is a main effect of PUFA source; B is a main effect of betaine; P*B is a significant effect of the interaction of betaine and polyunsaturated fat source; P and B appear when the f test showed a significance *p* ≤ 0.05.

**Table 5 animals-12-02837-t005:** Methylated glycines and sulfur amino acid metabolites which changed during the study while consuming control, or betaine, flax, and fish oil supplemented foods *.

Metabolite Class	Biochemical Name	Control	Control + Flax	Control + Fish Oil	Control + Betaine	Control + Flax + Betaine	Control + Fish Oil + Betaine
Methylated glycines	sarcosine	1.18	1.15	1.26	1.52	1.65	1.39
	dimethylglycine	1	1.14	1.13	1.75	2.29	1.53
	betaine	1.13	0.91	0.92	2.97	2.14	3.44
Methionine	methionine	1.11	1.08	1.15	1.26	1.52	1.2
	methionine sulfone	0.94	1.04	1.15	1.35	1.24	0.99
	methionine sulfoxide	1.05	0.92	1.06	1.24	1.58	1.31
	S-adenosylhomocysteine	1.01	0.92	0.92	1.14	1.53	0.87
	2,3-dihydroxy-5-methylthio-4-pentenoate	1.11	1.07	1.13	1.31	1.37	1.15
Cysteine	homocystine	1.37	5.69	1.68	1.83	5.06	0.94
	cystathionine	0.79	2.09	0.8	0.98	2.65	1.03
	alpha-ketobutyrate	0.75	1.16	0.98	0.97	0.76	0.54
	cysteine	1.21	1.31	1.23	1.31	1.45	0.96
	S-methylcysteine	0.98	1.14	0.97	1.04	1.12	1.14
	S-methylcysteine sulfoxide	0.9	1.34	0.94	0.95	1.28	1
	cysteine s-sulfate	1.31	1.4	1.35	1.12	0.99	0.99
	cystine	1.21	1.11	1.22	1.03	1.29	1.26
	lanthionine	0.95	1.77	0.94	1.07	2.28	1.15
	S-methylglutathione	0.91	1.04	1.22	1.09	1.36	1.14
	cysteinylglycine	1.23	1.46	1.43	1.25	1.44	0.79
	cysteinylglycine disulfide	1.34	1.23	1.36	1.23	1.21	1.32
Felinine	felinine	1.07	1.22	1.3	1.1	1.01	1.09
	gamma-glutamylfelinylglycine	0.99	1.05	1.25	0.92	1.03	1.04
	N-acetylfelinine	0.97	1.01	1.32	1.03	1	1.17
	felinylglycine	1.12	1.13	1.52	0.94	0.98	1.13

* This table consists of all analytes that had a *p* ≤ 0.05 and a q ≤ 0.1 for both change during the study and a difference between treatments at the end of the study. Values are change during the study while eating the listed food. Green denotes a decline and red an increase in the ratio (Final concentration/Initial concentration) of that compound.

**Table 6 animals-12-02837-t006:** Diacylglycerides and fatty acid dicarboxylates which changed during the study while consuming control, or betaine, flax, and fish oil supplemented foods *.

Metabolite Class	Biochemical	Control	Control + Flax	Control + Fish Oil	Control + Betaine	Control + Flax + Betaine	Control + Fish Oil + Betaine
Diacylglyceride	palmitoyl-oleoyl-glycerol (16:0/18:1)	1.37	0.56	1.68	1.48	1.86	1.26
	oleoyl-oleoyl-glycerol (18:1/18:1) [1]	1.42	0.96	1.33	0.97	1.77	0.88
	linoleoyl-linolenoyl-glycerol (18:2/18:3) [2]	0.89	1.24	0.86	0.67	1.41	0.64
	stearoyl-arachidonoyl-glycerol (18:0/20:4) [1]	1.08	0.99	0.98	1.31	1.2	0.96
	stearoyl-arachidonoyl-glycerol (18:0/20:4) [2]	1.12	1	1	1.3	1.33	1.04
	oleoyl-arachidonoyl-glycerol (18:1/20:4) [2]	1.15	1.07	1.26	1.21	1.17	1.04
	linoleoyl-arachidonoyl-glycerol (18:2/20:4) [2]	1.08	1.06	1.7	0.96	1.14	1.57
Dicarboxylate	tiglylcarnitine (C5:1-DC)	1.15	1.24	1.02	0.85	0.83	0.69
	pimeloylcarnitine/3-methyladipoylcarnitine (C7-DC)	0.87	1.07	1.09	0.49	0.73	0.48
	suberoylcarnitine (C8-DC)	0.96	1.06	0.94	0.55	0.56	0.76
	dodecanedioate (C12-DC)	0.9	0.94	0.99	0.87	0.99	0.82
	hexadecanedioate (C16-DC)	1.18	1.47	1.5	1.17	1.21	1.01
	octadecadienedioate (C18:2-DC)	0.98	0.93	0.93	1.12	1.17	0.95
	nonadecanedioate (C19-DC)	0.95	1.03	0.83	0.84	0.85	0.79
	eicosanedioate (C20-DC)	0.97	1.05	0.88	0.88	0.85	0.82
	docosadioate (C22-DC)	0.89	0.97	0.8	0.73	0.83	0.82

* This table consists of all analytes that had a *p* ≤ 0.05 and a q ≤ 0.1 for both change during the study and a difference between treatments at the end of the study. Values are change during the study while eating the listed food. Green denotes a decline and red an increase in the ratio (Final concentration/Initial concentration) of that compound.

**Table 7 animals-12-02837-t007:** Acylcarnitines which changed during the study while consuming control, or betaine, flax, and fish oil supplemented foods *.

Metabolite Class	Biochemical	Control	Control + Flax	Control + Fish Oil	Control + Betaine	Control + Flax + Betaine	Control + Fish Oil + Betaine
Precursor	carnitine	1.08	1.02	0.91	1.09	0.89	0.9
Short and medium chain acylcarnitines	acetylcarnitine (C2)	1.18	1.04	0.83	1.13	0.88	0.85
	butyrylcarnitine (C4)	1.18	1.01	0.9	0.84	0.85	0.69
	propionylcarnitine (C3)	1.22	0.99	0.86	0.77	0.81	0.75
	hexanoylcarnitine (C6)	1.3	1.08	0.78	0.85	0.73	0.76
	octanoylcarnitine (C8)	1.21	1.12	0.72	0.78	0.76	0.84
	decanoylcarnitine (C10)	1.23	1.18	0.79	0.84	0.76	0.81
	cis-4-decenoylcarnitine (C10:1)	1.19	0.96	0.69	0.81	0.72	0.75
Intermediate chain acylcarnitines	laurylcarnitine (C12)	1.03	1.08	0.88	0.76	0.72	0.72
	5-dodecenoylcarnitine (C12:1)	1.13	1.22	1.11	0.98	0.81	0.76
	myristoylcarnitine (C14)	1.08	1.25	1.02	0.9	0.87	0.75
	pentadecanoylcarnitine (C15)	1.18	1.03	2.26	0.89	0.71	0.8
Long chain acylcarnitines	palmitoylcarnitine (C16)	1.15	1.26	1.15	1.04	1.09	0.82
	palmitoleoylcarnitine (C16:1)	1.07	1.27	1.17	1.01	0.99	0.82
	margaroylcarnitine (C17)	1.18	1.38	1.24	1.02	0.93	0.82
	stearoylcarnitine (C18)	1.21	1.32	1.21	1.18	1.09	0.88
	oleoylcarnitine (C18:1)	1.18	1.31	1.08	1.07	1.13	0.83
	linoleoylcarnitine (C18:2)	1.18	1.33	1	0.99	1.03	0.76
	linolenoylcarnitine (C18:3)	0.88	1.34	0.75	0.64	0.94	0.6
Very long chain acylcarnitines	eicosenoylcarnitine (C20:1)	1.21	1.25	1.1	1.09	1.03	0.9
	dihomo-linoleoylcarnitine (C20:2)	1.23	1.29	1.03	1.12	1.12	0.81
	dihomo-linolenoylcarnitine (C20:3n3 or 6)	1.12	1.22	1.23	1.02	1.09	1.33
	arachidonoylcarnitine (C20:4)	1.2	1.32	1.02	1.07	1	0.87
	adrenoylcarnitine (C22:4)	1.3	1.22	0.84	1.02	1.05	0.68
	docosapentaenoylcarnitine (C22:5n3)	1.02	0.97	2.61	0.89	0.89	2.25
	docosahexaenoylcarnitine (C22:6)	0.95	0.72	5.15	0.67	0.71	5.2
	ximenoylcarnitine (C26:1)	1.04	1	0.91	0.69	0.78	0.73

* This table consists of all analytes that had a *p* ≤ 0.05 and a q ≤ 0.1 for both change during the study and a difference between treatments at the end of the study. Values are change during the study while eating the listed food. Green denotes a decline and red an increase in the ratio (Final concentration/Initial concentration) of that compound.

**Table 8 animals-12-02837-t008:** Choline-containing phospholipids which changed during the study while consuming control, or betaine, flax, and fish oil supplemented foods *.

Metabolite Class	Biochemical	Control	Control + Flax	Control + Fish Oil	Control + Betaine	Control + Flax + Betaine	Control + Fish Oil + Betaine
Precursor	choline phosphate	0.98	0.86	0.96	0.89	1.02	0.86
	glycerophosphorylcholine (GPC)	1.12	0.99	1.06	1.14	1.17	1.1
Choline-containing phospholipid	1-myristoyl-2-palmitoyl-GPC (14:0/16:0)	0.96	0.93	0.83	0.92	0.93	0.77
	1,2-dipalmitoyl-GPC (16:0/16:0)	0.92	0.86	1.13	1	1.02	1.27
	1-palmitoyl-2-palmitoleoyl-GPC (16:0/16:1)	1.02	0.97	0.78	1.07	1	0.66
	1-palmitoyl-2-stearoyl-GPC (16:0/18:0)	0.81	0.77	1.11	1.08	1.11	1.57
	1-palmitoyl-2-linoleoyl-GPC (16:0/18:2)	1.02	0.98	0.81	1.06	1.01	0.88
	1-palmitoyl-2-alpha-linolenoyl-GPC (16:0/18:3n3)	0.87	1.26	0.48	0.94	1.41	0.35
	1-palmitoyl-2-arachidonoyl-GPC (16:0/20:4n6)	1.02	0.98	0.94	1.13	1.08	1.02
	1-palmitoyl-2-docosahexaenoyl-GPC (16:0/22:6)	0.89	0.81	2.97	1.02	0.85	3.23
	1-palmitoleoyl-2-linoleoyl-GPC (16:1/18:2)	1.24	1.08	0.85	1.18	1.05	0.83
	1-palmitoleoyl-2-linolenoyl-GPC (16:1/18:3)	0.97	1.25	0.75	0.99	1.31	0.56
	1,2-distearoyl-GPC (18:0/18:0)	0.98	1.29	1.42	0.91	1.23	0.73
	1-stearoyl-2-oleoyl-GPC (18:0/18:1)	1.06	1.07	1.07	1.14	1.09	0.96
	1-stearoyl-2-linoleoyl-GPC (18:0/18:2)	1.03	1.03	0.93	1.06	1.07	0.99
	1-stearoyl-2-arachidonoyl-GPC (18:0/20:4)	1.01	1.04	1.02	1.11	1.11	1.02
	1-stearoyl-2-docosahexaenoyl-GPC (18:0/22:6)	0.86	0.81	3.05	0.94	0.89	3.26
	1-oleoyl-2-linoleoyl-GPC (18:1/18:2)	0.96	1.06	1.17	1.13	1.09	1.16
	1-oleoyl-2-docosahexaenoyl-GPC (18:1/22:6)	0.98	0.97	2.64	1.18	1	2.48
	1,2-dilinoleoyl-GPC (18:2/18:2)	0.99	1.03	0.66	1.15	1.21	0.7
	1-linoleoyl-2-linolenoyl-GPC (18:2/18:3)	0.88	1.34	0.6	0.92	1.56	0.6
	1-linoleoyl-2-arachidonoyl-GPC (18:2/20:4n6)	1.03	1.09	1.16	1.29	1.29	1.02
	1,2-dilinolenoyl-GPC (18:3/18:3)	0.55	1.19	1.3	0.59	1.48	0.7
Lysophospholipid	1-palmitoyl-GPC (16:0)	1.06	1.06	1.09	1.1	1.14	1.13
	1-palmitoleoyl-GPC (16:1)	1.17	1.11	1.07	1.21	1.25	0.9
	1-stearoyl-GPC (18:0)	1.08	1.12	1.15	1.15	1.22	1.16
	1-oleoyl-GPC (18:1)	1.05	1.04	1.03	1.26	1.28	1.05
	1-linoleoyl-GPC (18:2)	1	0.96	0.83	1.19	1.24	1
	1-linolenoyl-GPC (18:3)	0.86	1.31	0.74	0.91	1.75	0.63
	1-arachidonoyl-GPC (20:4n6)	1.04	1.07	1.07	1.37	1.41	1.2
	1-lignoceroyl-GPC (24:0)	0.95	0.97	1.08	1.1	1.1	1.16
	1-cerotoyl-GPC (26:0)	0.93	1.18	0.6	0.98	0.98	0.72
Choline-containing plasmalogen	1-(1-enyl-palmitoyl)-2-palmitoyl-GPC (P-16:0/16:0)	0.95	0.93	1.1	0.94	0.77	1.11
	1-(1-enyl-palmitoyl)-2-palmitoleoyl-GPC (P-16:0/16:1)	0.82	1.04	0.6	0.95	0.96	0.67
	1-(1-enyl-palmitoyl)-2-oleoyl-GPC (P-16:0/18:1)	0.97	0.97	1.2	1	0.85	1.34
	1-(1-enyl-palmitoyl)-2-arachidonoyl-GPC (P-16:0/20:4)	0.97	1.03	0.8	0.96	0.97	1.01
	1-(1-enyl-palmitoyl)-2-linoleoyl-GPC (P-16:0/18:2)	1.09	1.09	0.65	1.18	1.08	0.66
Lysoplasmalogen	1-(1-enyl-palmitoyl)-GPC (P-16:0)	1.01	1.08	1.13	1.1	1.18	1.2

* This table consists of all analytes that had a *p* ≤ 0.05 and a q ≤ 0.1 for both change during the study and a difference between treatments at the end of the study. Values are change during the study while eating the listed food. Green denotes a decline and red an increase in the ratio (Final concentration/Initial concentration) of that compound.

**Table 9 animals-12-02837-t009:** Ethanolamine- and inositol-containing phospholipids which changed during the study while consuming control, or betaine, flax, and fish oil supplemented foods *.

Metabolite Class	Biochemical	Control	Control + Flax	Control + Fish Oil	Control + Betaine	Control + Flax + Betaine	Control + Fish Oil + Betaine
Ethanolamine-containing phospholipid	1-palmitoyl-2-oleoyl-GPE (16:0/18:1)	1.23	1	1	1.46	1.32	0.74
	1-palmitoyl-2-arachidonoyl-GPE (16:0/20:4)	1.08	1	0.9	1.13	1.01	0.7
	1-palmitoyl-2-docosahexaenoyl-GPE (16:0/22:6)	0.96	1	3	1.1	0.86	2.24
	1-stearoyl-2-linoleoyl-GPE (18:0/18:2)	1.07	0.9	0.6	1.1	1.15	0.66
	1-stearoyl-2-arachidonoyl-GPE (18:0/20:4)	1.03	1.1	0.9	1.06	1.03	0.72
	1-stearoyl-2-docosahexaenoyl-GPE (18:0/22:6)	0.98	0.9	1.7	0.68	0.76	1.6
	1-oleoyl-2-linoleoyl-GPE (18:1/18:2)	1.01	0.6	0.3	0.87	0.98	0.34
	1,2-dilinoleoyl-GPE (18:2/18:2)	1.08	1.1	0.5	1.13	1.28	0.44
	1-linoleoyl-2-arachidonoyl-GPE (18:2/20:4)	1.22	1.1	0.6	1.27	1.13	0.44
Lysophospholipid	1-palmitoyl-GPE (16:0)	1.04	1	1.6	0.97	1.08	1.37
	1-stearoyl-GPE (18:0)	1.05	1.1	1.2	1.05	1.1	1.05
	1-oleoyl-GPE (18:1)	1.2	1	0.9	1.5	1.43	0.72
	1-linoleoyl-GPE (18:2)	1.15	1.1	0.7	1.26	1.21	0.64
	1-arachidonoyl-GPE (20:4n6)	1.2	1.2	0.9	1.18	1.1	0.73
Ethanolamine-containing plasmalogen	1-(1-enyl-palmitoyl)-2-oleoyl-GPE (P-16:0/18:1)	1	0.9	0.8	1.04	1.1	0.8
	1-(1-enyl-palmitoyl)-2-arachidonoyl-GPE (P-16:0/20:4)	0.93	1	0.8	0.97	0.87	0.78
	1-(1-enyl-stearoyl)-2-linoleoyl-GPE (P-18:0/18:2)	0.95	0.9	0.8	1.15	1.09	1.03
	1-(1-enyl-stearoyl)-2-arachidonoyl-GPE (P-18:0/20:4)	1.1	1.1	0.9	1.21	1.16	1.12
Lysoplasmalogen	1-(1-enyl-palmitoyl)-GPE (P-16:0)	1.02	1.1	1.2	1.12	1.07	1.12
	1-(1-enyl-oleoyl)-GPE (P-18:1)	1.1	1.1	1.4	1.16	1.26	1.05
	1-(1-enyl-stearoyl)-GPE (P-18:0)	1.09	1.1	1.3	1.24	1.22	1.25
Inositol-containing phospholipid	1-stearoyl-2-oleoyl-GPI (18:0/18:1)	1.07	1	1.2	1.14	1.27	1.18
	1-stearoyl-2-linoleoyl-GPI (18:0/18:2)	1.03	1	1	1.12	1.22	1.09
	1-palmitoyl-2-arachidonoyl-GPI (16:0/20:4)	1.04	0.9	1	1.03	1.14	1.26
	1-stearoyl-2-arachidonoyl-GPI (18:0/20:4)	1.02	1	1.1	1.1	1.18	1.15
	1-oleoyl-2-arachidonoyl-GPI (18:1/20:4)	1.18	1.1	0.9	1.19	1.26	0.79

* This table consists of all analytes that had a *p* ≤ 0.05 and a q ≤ 0.1 for both change during the study and a difference between treatments at the end of the study. Values are change during the study while eating the listed food. Green denotes a decline and red an increase in the ratio (Final concentration/Initial concentration) of that compound.

**Table 10 animals-12-02837-t010:** Sphingolipids which changed during the study while consuming control, or betaine, flax, and fish oil supplemented foods *.

Metabolite Class	Biochemical	Control	Control + Flax	Control + Fish Oil	Control + Betaine	Control + Flax + Betaine	Control + Fish Oil + Betaine
Precursor	sphinganine	1.96	1.12	1.32	1.87	1.99	1.95
	sphinganine-1-phosphate	1.48	1.13	1.23	1.15	1.32	1.17
	sphingadienine	2.3	1.07	1.16	2.08	2.55	1.79
	sphingosine	2.05	1.19	1.36	1.91	1.92	1.96
	sphingosine 1-phosphate	1.31	1.14	1.24	1.13	1.18	1.13
Sphingolipid	sphingomyelin (d17:1/16:0, d18:1/15:0, d16:1/17:0)	1.05	1.01	0.98	1.11	1.13	1.13
	sphingomyelin (d17:2/16:0, d18:2/15:0)	1.05	1	1.12	1.2	1.2	1.33
	N-palmitoyl-sphinganine (d18:0/16:0)	1.03	0.88	0.99	1.39	1.52	1.43
	N-stearoyl-sphinganine (d18:0/18:0)	1.16	1.03	1.24	1.47	1.53	1.56
	N-palmitoyl-sphingosine (d18:1/16:0)	1.01	0.95	1.03	1.26	1.25	1.24
	N-stearoyl-sphingosine (d18:1/18:0)	1.05	1.01	1.12	1.29	1.28	1.3
	N-palmitoyl-sphingadienine (d18:2/16:0)	1.11	1.1	0.94	1.08	1.09	1.17
	N-behenoyl-sphingadienine (d18:2/22:0)	0.97	0.97	1.15	1.48	1.44	1.66
	N-palmitoyl-heptadecasphingosine (d17:1/16:0)	1	1	0.97	1.14	1.23	1.21
	ceramide (d18:1/14:0, d16:1/16:0)	0.91	0.83	0.8	0.89	1.02	1.01
	ceramide (d18:1/17:0, d17:1/18:0)	1.01	0.96	1.11	1.25	1.22	1.31
	ceramide (d18:1/20:0, d16:1/22:0, d20:1/18:0)	1.02	1.01	1.13	1.26	1.33	1.18
	ceramide (d18:2/24:1, d18:1/24:2)	1.07	0.99	1.1	1.11	1.27	1.23
	glycosyl-N-palmitoyl-sphingosine (d18:1/16:0)	0.96	0.95	1.09	1.16	1.12	1.31
	glycosyl-N-stearoyl-sphingosine (d18:1/18:0)	1.05	1.04	1.21	1.21	1.17	1.19
	glycosyl-N-arachidoyl-sphingosine (d18:1/20:0)	1.05	1.06	1.23	1.21	1.12	1.26
	glycosyl-N-(2-hydroxynervonoyl)-sphingosine (d18:1/24:1(2OH))	1.08	0.88	1.27	1.12	1.26	1.28
	glycosyl ceramide (d18:2/24:1, d18:1/24:2)	1.1	1.1	1.21	1.26	1.24	1.33
	lactosyl-N-palmitoyl-sphingosine (d18:1/16:0)	1.11	1.11	1.13	1.22	1.2	1.22
	palmitoyl dihydrosphingomyelin (d18:0/16:0)	0.89	0.95	1.11	1.09	1.21	1.38
	behenoyl dihydrosphingomyelin (d18:0/22:0)	0.95	0.99	1.37	1.2	1.47	1.3
	sphingomyelin (d18:0/18:0, d19:0/17:0)	0.95	1.02	1.22	1.07	1.28	1.51
	sphingomyelin (d18:0/20:0, d16:0/22:0)	0.98	0.98	1.43	1.14	1.32	1.35
	palmitoyl sphingomyelin (d18:1/16:0)	1	1.03	0.9	1.08	1.08	1
	hydroxypalmitoyl sphingomyelin (d18:1/16:0(OH))	1.02	1.07	1.07	1.06	1.12	1.2
	stearoyl sphingomyelin (d18:1/18:0)	1	1.01	1.11	0.98	0.99	1.22
	behenoyl sphingomyelin (d18:1/22:0)	1.07	1.07	1.11	1.1	1.13	1.17
	tricosanoyl sphingomyelin (d18:1/23:0)	1.09	1.07	1.16	1.18	1.2	1.23
	lignoceroyl sphingomyelin (d18:1/24:0)	1.09	1.05	1.18	1.21	1.24	1.24
	sphingomyelin (d18:2/18:1)	1.07	1.05	0.87	1.13	1.2	1.04
	sphingomyelin (d18:2/23:1)	1.05	0.96	1.1	0.97	1.02	1.34
	sphingomyelin (d18:1/14:0, d16:1/16:0)	0.91	0.89	0.82	0.92	0.98	0.94
	sphingomyelin (d18:2/14:0, d18:1/14:1)	1.02	1.02	1.15	1.06	1.23	1.16
	sphingomyelin (d18:2/16:0, d18:1/16:1)	1.06	1.06	0.92	1.19	1.16	1.1
	sphingomyelin (d18:1/17:0, d17:1/18:0, d19:1/16:0)	1.06	1.05	1.09	1.09	1.08	1.16
	sphingomyelin (d18:1/18:1, d18:2/18:0)	1.08	1.09	0.94	1.14	1.09	1.02
	sphingomyelin (d18:1/19:0, d19:1/18:0)	1.03	1	1.08	1.04	1.03	1.18
	sphingomyelin (d18:1/20:0, d16:1/22:0)	1.03	1.01	1.07	1.02	1.01	1.16
	sphingomyelin (d18:1/20:1, d18:2/20:0)	1.03	1.03	1.18	1	1.02	1.29
	sphingomyelin (d18:1/20:2, d18:2/20:1, d16:1/22:2)	1.02	1.03	0.71	0.8	0.95	0.74
	sphingomyelin (d18:2/21:0, d16:2/23:0)	1.07	0.96	1.1	1.03	1.02	1.24
	sphingomyelin (d18:1/22:1, d18:2/22:0, d16:1/24:1)	0.95	0.95	1.05	0.89	0.95	1.12
	sphingomyelin (d18:1/22:2, d18:2/22:1, d16:1/24:2)	0.96	0.93	0.89	0.81	0.9	1
	sphingomyelin (d18:2/23:0, d18:1/23:1, d17:1/24:1)	1.09	1.03	1.18	1.11	1.17	1.28
	sphingomyelin (d18:1/24:1, d18:2/24:0)	1.09	1.06	1.17	1.06	1.1	1.25
	sphingomyelin (d18:2/24:1, d18:1/24:2)	1.03	1.02	1.09	1.03	1.05	1.22
	sphingomyelin (d18:1/25:0, d19:0/24:1, d20:1/23:0, d19:1/24:0)	0.98	1.01	1.08	1.02	0.98	1.09
	palmitoyl-sphingosine-phosphoethanolamine (d18:1/16:0)	1.09	1.06	0.68	1.02	0.99	0.73

* This table consists of all analytes that had a *p* ≤ 0.05 and a q ≤ 0.1 for both change during the study and a difference between treatments at the end of the study. Values are change during the study while eating the listed food. Green denotes a decline and red an increase in the ratio (Final concentration/Initial concentration) of that compound.

**Table 11 animals-12-02837-t011:** Endocannabinoids and N-acylated amino acids which changed during the study while consuming control, or betaine, flax, and fish oil supplemented foods *.

Metabolite Class	Biochemical	Control	Control + Flax	Control + Fish Oil	Control + Betaine	Control + Flax + Betaine	Control + Fish Oil + Betaine
Ethanolamide	oleoyl ethanolamide	1.14	1.21	1.17	1.17	1.17	0.89
	palmitoyl ethanolamide	1.04	1.08	1.03	1.1	1.1	1.06
	stearoyl ethanolamide	1.03	1.07	1.06	1.15	1.14	1.04
Acyltaurine	N-myristoyltaurine	1.37	1.1	0.95	1.17	0.79	0.67
	N-oleoyltaurine	1.17	1.22	0.93	1.16	0.88	0.69
	N-palmitoyltaurine	1.46	1.48	0.94	1.42	0.82	0.76
	N-palmitoleoyltaurine	1.22	1.24	1.06	1.12	0.84	0.75
	N-linolenoyltaurine	1.12	1.28	0.62	0.98	0.99	0.54
Acylserine	N-stearoylserine	1.2	0.87	0.67	2.01	1.24	1.43
Acylglycine	N-palmitoylglycine	1.22	1.16	0.78	1.34	0.83	0.88

* This table consists of all analytes that had a *p* ≤ 0.05 and a q ≤ 0.1 for both change during the study and a difference between treatments at the end of the study. Values are change during the study while eating the listed food. Green denotes a decline and red an increase in the ratio (Final concentration/Initial concentration) of that compound.

**Table 12 animals-12-02837-t012:** Bile acids and cholesterol metabolites which changed during the study while consuming control, or betaine, flax, and fish oil supplemented foods *.

Metabolite Class	Biochemical	Control	Control + Flax	Control + Fish Oil	Control +Betaine	Control + Flax + Betaine	Control + Fish Oil + Betaine
Bile Acids	3beta-hydroxy-5-cholestenoate	1.6	1.56	0.57	1.56	1.05	0.56
	7-alpha-hydroxy-3-oxo-4-cholestenoate (7-Hoca)	1.17	1.16	0.88	0.96	1.09	0.81
	deoxycholate	1.2	0.69	0.56	1.31	1.22	0.95
	taurocholenate sulfate	1.07	1.06	0.73	1.32	1.12	0.93
Cholesterol	3-hydroxy-3-methylglutarate	1.03	0.99	0.98	1.14	1.01	0.94
	mevalonate	1.2	0.97	1.08	0.69	0.83	0.62
	cholesterol	1.1	1.02	1.09	1.21	1.3	1.25

* This table consists of all analytes that had a *p* ≤ 0.05 and a q ≤ 0.1 for both change during the study and a difference between treatments at the end of the study. Values are change during the study while eating the listed food. Green denotes a decline and red an increase in the ratio (Final concentration/Initial concentration) of that compound.

## Data Availability

Data are available either in the manuscript or from the corresponding author.

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
