# Peer review of "Dietary Betaine Interacts with Very Long Chain n-3 Polyunsaturated Fatty Acids to Influence Fat Metabolism and Circulating Single Carbon Status in the Cat"

_animals, 2022, doi:10.3390/ani12202837_

Round 1

Reviewer 1 Report

This study sought to understand how intake of dietary PUFA and betaine affected circulating PUFA in cats. The data are interesting and the manuscript is important to our understanding of the effects of diet on physiological responses and my comments are to build on the paper. While I feel strongly that this is important data, I do think that the statistics may need to be re-visited.

Simple summary: I would suggest a more pointed simple summary that provides recommendations for PUFA intake and betaine intake to maximize the health of cats.

Abstract:

  • While your diet design is a factorial design, but it is not clear how you allocated cats and analyzed the data. Please add a statement with regards to this.
  • Were the cats fed to weight maintenance, loss or gain?

Introduction:

  • Please add hypotheses regarding the study
  • I think that the final sentence is misleading and contributing to increasing and lengthening the health and longevity is not an outcome of the present study. You can discuss that supporting health throughout life could lead to enhanced longevity, but putting it in the response criteria is a bit misleading.
  • - Your rationale for PUFA oil sources and betaine are solid, but why you would combine them lacks rationale. Please provide further information to support the experimental design.

Materials and methods:

Line 95: “masked” should be “blinded”

  • Please provide the characteristics of the cats used in the study (sex split, average BW, BCS and age) and the means per treatment group.
  • Please provide the wash in diet as its composition predicts your baseline. 
  • Diet concentrations and your diet table should be presented in materials and methods as treatment is an independent variable that you control, it should not be presented in results. Please consider moving this table, which compliments your text on diet composition. Furthermore, how you processed your diets (did you use the batch ingredients, were the processing conditions similar) need to be added. Next, please indicate how you analyzed the nutrients in the diets and how many replicates you had in your analyses. Finally, because you have choline and carnitine derivatives, should add you add dietary choline and carnitine content too? Choline in the form of choline-chloride vs. Choline as phophatidylcholine should be considered too.
  • Line 109: This is not sufficient information regarding your blood sample. What method for blood sampling was employed, how much blood was taken, were cats sedated and if no, did you evaluate stress during sampling?
  • Line 115: It is not clear whether you are just looking at free fatty acids or fatty acid composition of the lipid fractions. Please clarify.
  • Line 118-120: This ratio of circulating FA to dietary intake, this is a simplistic view of metabolism. You provide no rationale for this calculation or validation of what that calculation means. If it is not validated, then remove. If it is validated then provide information regarding the use of this calculation.
  • Line 122-127: This should be moved up in your materials and methods prior to the description of the conduct of the study. More information is required regarding housing. Were all cats housed in a single room? Provide details as to the room size, number of cats etc. If multiple rooms were used, did you consider this in your statistics? Furthermore, how did you feed treatment diets? Group housed cats are a challenge and it is unclear how these cats are individually fed, how intake was monitored etc.
  • Line 125: Please provide the calculations you used to calculate maintenance food intake or a better explanation, such as basing the ME intake on historical feeding and body weight records.
  • Statistics: Why use such an antiquated procedure in SAS? A Mixed model or GLIMMIX would allow you to also account for repeated measures and for the individual effect of “cat” as your subject. Thank you for ensuring you have methods for FDR. I understand how you identified when means are different, but what multiple comparison did you use to separate your means?

Results:

Line 139-145: This information should go in your materials and methods, diet is a fixed effect and independent variables, so M&M. 

Table 2: You do state the effect of diet on cholesterol, but you do not present any other P values. This is unusual to not provide P values within the table. The non-significant results also need to be written out.

Line 270-272: Be careful using “sparing” as this refers to one dietary nutrient being able to replace another. Classically, Tyrosine for Phenylalanine and Cysteine/Cystine for Methionine. I would not use “spare” for choline and betaine as of yet. It does alter intermediary metabolism, but it is not clear if you would get the biological responses using choline and betaine.

  • It is my opinion that you should provide P values for diet, and added oil in your tables.
  •  

Discussion:

Line 403: At the end of this paragraph I am left wondering about how the ratio of N6: N3 along with betaine (or presumably choline as well) alter the inflammatory profile. Any take home message for the CON vs. FLAX and FISH?

Paragraph Line 405-423: There is a lot of work with added choline in the cat recently and curious for you to comment on similarities and differences with choline supplementation rather than betaine? This is an especially important comparison as you suggest that betaine can help to increase the availability of choline.

Conflicts of interest: There is increased scrutiny of companion animal nutritionists with industrial ties, including funding. You will have to provide statements regarding whether the funder, Hills Pet Nutrition, played any role or had any decision making about what was allowed to be included in the report.

Author Response

This study sought to understand how intake of dietary PUFA and betaine affected circulating PUFA in cats. The data are interesting and the manuscript is important to our understanding of the effects of diet on physiological responses and my comments are to build on the paper. While I feel strongly that this is important data, I do think that the statistics may need to be re-visited.

Simple summary: I would suggest a more pointed simple summary that provides recommendations for PUFA intake and betaine intake to maximize the health of cats.

We have reworded the summary and added a PUFA and betaine intake recommendation

Abstract:

  • While your diet design is a factorial design, but it is not clear how you allocated cats and analyzed the data. Please add a statement with regards to this.

A statement was added for clarification

  • Were the cats fed to weight maintenance, loss or gain?

A statement was added for clarification

Introduction:

  • Please add hypotheses regarding the study
    A statement was added for clarification
  • I think that the final sentence is misleading and contributing to increasing and lengthening the health and longevity is not an outcome of the present study. You can discuss that supporting health throughout life could lead to enhanced longevity, but putting it in the response criteria is a bit misleading.
  • A change in wording was done and this aspirational thought was deleted
  • - Your rationale for PUFA oil sources and betaine are solid, but why you would combine them lacks rationale. Please provide further information to support the experimental design.
    A statement was added for clarification

Materials and methods:

Line 95: “masked” should be “blinded”

We disagree, blinded has the unfortunate connotation of physical harm and masked is preferred.

  • Please provide the characteristics of the cats used in the study (sex split, average BW, BCS and age) and the means per treatment group.
    The means of average BW are shown in table 2, sex split and age were added. We did not measure or estimate body condition.
  • Please provide the wash in diet as its composition predicts your baseline. 

Control food was used as a wash in food and that information is now redundantly stated for clarity.

  • Diet concentrations and your diet table should be presented in materials and methods as treatment is an independent variable that you control, it should not be presented in results. Please consider moving this table, which compliments your text on diet composition. Furthermore, how you processed your diets (did you use the batch ingredients, were the processing conditions similar) need to be added. Next, please indicate how you analyzed the nutrients in the diets and how many replicates you had in your analyses. Finally, because you have choline and carnitine derivatives, should add you add dietary choline and carnitine content too? Choline in the form of choline-chloride vs. Choline as phophatidylcholine should be considered too.

This information was added for clarity.

  • Line 109: This is not sufficient information regarding your blood sample. What method for blood sampling was employed, how much blood was taken, were cats sedated and if no, did you evaluate stress during sampling?
    A statement was added for clarification
  • Line 115: It is not clear whether you are just looking at free fatty acids or fatty acid composition of the lipid fractions. Please clarify.

Assay information was added, it was complete quantification of the lipid fractions.

  • Line 118-120: This ratio of circulating FA to dietary intake, this is a simplistic view of metabolism. You provide no rationale for this calculation or validation of what that calculation means. If it is not validated, then remove. If it is validated then provide information regarding the use of this calculation.

This is an informational outcome using validated measures.  It may in fact be simplistic but it is quite informative. Note for example the response of DHA to treatment. The cat does not construct from DHA from αLA. This leads us to the conclusion that DHA is not responding through elongation and desaturation from αLA.  DHA concentration is not statistically increased in response to dietary betaine.  Although it is numerically higher with betaine added.  However, when expressed as a ratio there is an interaction with PUFA and betaine this shows the benefit of the calculation because it reduces the within treatment variation by taking into account the change in circulation associated with change in intake.  This is added to the materials and methods for the reader to understand.

  • Line 122-127: This should be moved up in your materials and methods prior to the description of the conduct of the study. More information is required regarding housing. Were all cats housed in a single room? Provide details as to the room size, number of cats etc. If multiple rooms were used, did you consider this in your statistics? Furthermore, how did you feed treatment diets? Group housed cats are a challenge and it is unclear how these cats are individually fed, how intake was monitored etc.

They are group housed and the electronic feeders control access and record consumption.  This information is expanded in the materials and methods for clarity.

  • Line 125: Please provide the calculations you used to calculate maintenance food intake or a better explanation, such as basing the ME intake on historical feeding and body weight records.

Information was added for clarity (food intake calories were previously established which maintained weight)

  • Statistics: Why use such an antiquated procedure in SAS? A Mixed model or GLIMMIX would allow you to also account for repeated measures and for the individual effect of “cat” as your subject. Thank you for ensuring you have methods for FDR. I understand how you identified when means are different, but what multiple comparison did you use to separate your means?

I used the GLM procedure because it gives a similar answer to the effect of the foods as do PROC GLIMMIX or MIXED.  It makes a better response graph for my personal use (these aren’t publication quality but are informative) so I use it when the more modern analysis gives similar result.  Note that the response variables for the fatty acids use initial values as a covariant which brings in the value of that initial concentration.  Some of the responses in the study aren’t repeated measures but rather individual responses such as the delta evaluation of fatty acid change over the study.  Mean separation method for the SAS output was added.

Results:

Line 139-145: This information should go in your materials and methods, diet is a fixed effect and independent variables, so M&M. 

Table 2: You do state the effect of diet on cholesterol, but you do not present any other P values. This is unusual to not provide P value     s within the table. The non-significant results also need to be written out.

The non-significant results were added in written form and the p value updated to the specific comparison.  As this was the only changed response variable it was an improvement.  We did not use each of the p values in the subsequent table as it reduces readability.

Line 270-272: Be careful using “sparing” as this refers to one dietary nutrient being able to replace another. Classically, Tyrosine for Phenylalanine and Cysteine/Cystine for Methionine. I would not use “spare” for choline and betaine as of yet. It does alter intermediary metabolism, but it is not clear if you would get the biological responses using choline and betaine.

I agree, wording was changed to not describe betaine as sparing choline as that was not tested and may in fact be untrue in the cat.

Wording

  • It is my opinion that you should provide P values for diet, and added oil in your tables.

A statement was added for clarification.  We continue to maintain that the table becomes less readable and the pre-chosen probability cutoff is the best way of reporting the differences.

Discussion:

Line 403: At the end of this paragraph I am left wondering about how the ratio of N6: N3 along with betaine (or presumably choline as well) alter the inflammatory profile. Any take home message for the CON vs. FLAX and FISH?

I think the only message is that there is an indication of a benefit and more work is needed to understand it – this was added for enhancement of the discussion.

Paragraph Line 405-423: There is a lot of work with added choline in the cat recently and curious for you to comment on similarities and differences with choline supplementation rather than betaine? This is an especially important comparison as you suggest that betaine can help to increase the availability of choline.

There is an expectation of similar responses and yet that’s as far as this work can go as they were not varied together.  We have added this idea to communicate that to the reader.

Conflicts of interest: There is increased scrutiny of companion animal nutritionists with industrial ties, including funding. You will have to provide statements regarding whether the funder, Hills Pet Nutrition, played any role or had any decision making about what was allowed to be included in the report.

Reviewer 2 Report

Specific comments

Lines 35-37: This observation is quite interesting, but relevant data was not included in the manuscript, and no discussion of this in the “discussion” section either. So relevant data and discussion should be included in the manuscript before it is accepted for publication.

Please provide baseline information of ages, gender distribution, body weight and BCS of the cats in each dietary groups

How were the cats fed? Were they fed ad libitum?

Table 1: why did the fish oil plus betaine diet contain 600-800 less betaine compared to other betaine-supplemented diets? Why did the flax control contain less αLA compared with flax control plus betaine?

Table 2:  Cats consuming betaine showed a higher concentration of cholesterol compared with the unsupplemented controls. But it seems that there may be significant difference in cholesterol at baseline between flax control and flax control plus betaine, between fish oil control and fish oil control pluas betaine? It is more appropriate to compare baseline and final cholesterol within each dietary group to determine whether the treatment significantly increased cholesterol from baseline.  

Table 3: why did the LA and αLA differ significantly at baseline even the cats were maintained on the control diet? This suggests that the 14-day washout was not long enough to standardized baseline LA and αLA, which could influence the treatment effects.

Usually adding DHA and EPA to a diet reduces ARA, why do you think that the fish oil diet increased ARA even without betaine supplementation in your study?

Table 4: Significant differences in multiple fatty acids at baseline again indicated that the 14-day washout was not long enough to make the circulating fatty acids comparable among 6 dietary groups, which confounds the observed treatment effects on oleic acid and palmitoleic acid.  

Did all six diets contain same protein sources? If so, please mentioned in the manuscript because this can affect the effects of the diets on blood amino acids.

Did you measure B vitamins in the diets, especially B6, B12 and folate? If so, please add the information to the manuscript.

Table 5: Why did homocysteine increase in five out of the 6 dietary groups except fish oil plus betaine group? Why did flax seed increase homocysteine so much? Please change “homocysteine” to “homocysteine”

Did you measure the blood B6, B12 and folate in those cats to determine whether the treatments affect the status of those B vitamins?

Why did fax control, fish oil control and control plus betaine increase felinine?

 Table 6. Why do you think that fish oil control, betaine control and flax see control plus betaine increased DAGs?

 Table 7: In table 6, three diets, fish oil control, betaine control and flax see control plus betaine, increased DAGs, suggesting that those diets increased lipolysis. In this table, only control and flax control increased acylcarnitines (CAN), indiacting those diets enhanced fatty acid oxidation in mitochondria. Why didn’t the data in tables 6 and 7 overlap each other?   what happened to the free fatty acids after the above three diets increased lipolysis?

 Table 8. Did you measure choline in six diets? If so, please add to the manuscript along with B6, B12 and folate data.

Author Response

Specific comments

Lines 35-37: This observation is quite interesting, but relevant data was not included in the manuscript, and no discussion of this in the “discussion” section either. So relevant data and discussion should be included in the manuscript before it is accepted for publication.

We have enhanced the discussion.  It’s relevant as in the dog we have previously published that betaine influenced the EPA concentration while flax addition resulted in an intermediate concentration (between control and added fish oil).  However, this paper shows that in the cat one sees a significant effect of betaine (and of course fish oil) on circulating DHA. As this study  cannot distinguish between any elongation or desaturation site the assignment of these changes is beyond its scope this was included in the discussion.

Please provide baseline information of ages, gender distribution, body weight and BCS of the cats in each dietary groups

This information was added although body weight by dietary group was already present and BCS was not measured.

How were the cats fed? Were they fed ad libitum?

M&M was enhanced to include this and its methodology (cats were fed ad libitum of a controlled amount to maintain body weight).

Table 1: why did the fish oil plus betaine diet contain 600-800 less betaine compared to other betaine-supplemented diets? Why did the flax control contain less αLA compared with flax control plus betaine?

These are single samples and subsequent analysis so the reported variation of similar dietary additions is a response to mixing, sampling, and analytical variation.

Table 2:  Cats consuming betaine showed a higher concentration of cholesterol compared with the unsupplemented controls. But it seems that there may be significant difference in cholesterol at baseline between flax control and flax control plus betaine, between fish oil control and fish oil control pluas betaine? It is more appropriate to compare baseline and final cholesterol within each dietary group to determine whether the treatment significantly increased cholesterol from baseline.  

This analysis was completed and added to the results (all of the treatment groups consuming dietary betaine had an increase).

Table 3: why did the LA and αLA differ significantly at baseline even the cats were maintained on the control diet? This suggests that the 14-day washout was not long enough to standardized baseline LA and αLA, which could influence the treatment effects.

Yes, we agree that one cannot establish why there were differences at baseline.  However, the controls were numerically declining while the cats consuming flax increased αLA.  Therefore,  the conclusion that dietary αLA was the driver of the increased circulation concentration is reasonable.

Usually adding DHA and EPA to a diet reduces ARA, why do you think that the fish oil diet increased ARA even without betaine supplementation in your study?

We agree that fish oil consumption normally reduces ARA so the slight increase in the non-betaine supplemented food is not the normal response.  However, the dietary increase of ARA with this fish oil may be the reason for this and the significant reduction in ARA from the food containing betaine suggests fish oil is having its normal response.

Table 4: Significant differences in multiple fatty acids at baseline again indicated that the 14-day washout was not long enough to make the circulating fatty acids comparable among 6 dietary groups, which confounds the observed treatment effects on oleic acid and palmitoleic acid.  

The paper’s conclusion that the differences at baseline do not describe the differences reported seems appropriate.  For example with oleic acid betaine increased the concentration similarly for the control and control+flax while in both the foods where fish oil was added there was a similar decline.  This supports the conclusion that the responses did not depend on initial concentration.

Did all six diets contain same protein sources? If so, please mentioned in the manuscript because this can affect the effects of the diets on blood amino acids.

All six foods contained the same protein sources at the same levels.  This information was added for clarity.

Did you measure B vitamins in the diets, especially B6, B12 and folate? If so, please add the information to the manuscript.

We did not measure these vitamins.  However, they were all added at the same level from the same premix and this information was added for clarity.

Table 5: Why did homocysteine increase in five out of the 6 dietary groups except fish oil plus betaine group? Why did flax seed increase homocysteine so much? Please change “homocysteine” to “homocysteine”

This was interesting and is now added to the discussion.  It seems most likely to be through an effect on the B6 activity due to an inhibitor in flax.

Did you measure the blood B6, B12 and folate in those cats to determine whether the treatments affect the status of those B vitamins?

Those vitamins were not analyzed

Why did fax control, fish oil control and control plus betaine increase felinine?

 Table 6. Why do you think that fish oil control, betaine control and flax see control plus betaine increased DAGs?

A possible mechanism was added to the discussion.

  Table 7: In table 6, three diets, fish oil control, betaine control and flax see control plus betaine, increased DAGs, suggesting that those diets increased lipolysis. In this table, only control and flax control increased acylcarnitines (CAN), indiacting those diets enhanced fatty acid oxidation in mitochondria. Why didn’t the data in tables 6 and 7 overlap each other?   what happened to the free fatty acids after the above three diets increased lipolysis?

We’ve added a discussion dealing with these changes.  However “I don’t know” would succinctly answer your question.

  Table 8. Did you measure choline in six diets? If so, please add to the manuscript along with B6, B12 and folate data.

These were not measured,  all foods were similarly supplemented and that was added for clarity.

Round 2

Reviewer 1 Report

Additional comments for the authors:

While the authors have added a statement regarding hypotheses, this is not a pointed hypothesis, this is very broad. Is there any reason why authors did not provide a hypotheses among treatments, specifically how n3 from fish vs. flax and with and without betaine? A hypothesis would be greater or less than or increased and decreased, stating that they will be different is not.

I understand that this is a study supported by Hills and likely a commercial formula, but stating that the foods had the same amount of “high concentration of high protein sources of chicken and corn gluten meal does not speak to the use of ingredients as the baseline. This is NOT sufficient and at minimum the ingredient deck of each formula needs to be added as a footnote. I agree that animals require nutrients rather than ingredients, but ingredients bring in different nutrients and the reader should know the relative compositions of these foods for a nutrition article. These must be added to allow the reader the opportunity to understand your dietary and as such, experimental treatments.

Line 128: Choline chloride is horribly hydrophobic, did you ensure sufficient mixing of choline chloride?

Carnitine, which you spelled incorrectly in your revised manuscript, is not available as L-carnitine for use in extruded food, usually it is added as carnitine tartrate, but the form is important to add. Further to this, the supplier of both the choline and carnitine needs to be stated.

Line 139: This is problematic that not all cats were sedated and you do not disclose whether you used that in your statistical model. Sedation and stress alter the metabolize, your outcome measure. Also, describing the sedation is important as well. Please address this. Sedation affects intermediary metabolism and that is what you are measuring. In the future, I would recommend sedating all cats as this would reduce the variability among cats and addresses the animal welfare aspects and the human health and safety while working with this animal. At minimum you would need to discuss this lack of experimental control of a significant variable as a major limitation to the study.

Housing and socialization was not addressed. The original comments was: “Line 122-127: This should be moved up in your materials and methods prior to the description of the conduct of the study. More information is required regarding housing. Were all cats housed in a single room? Provide details as to the room size, number of cats etc. If multiple rooms were used, did you consider this in your statistics? Furthermore, how did you feed treatment diets? Group housed cats are a challenge and it is unclear how these cats are individually fed, how intake was monitored etc.?”

     Your response is insufficient. You need to clearly define the housing and socialization that the cats received.  We know the cats are group housed, but nothing about how they were cared for and managed. This is important as  a role of the reviewer is to ensure that the research meets all applicable standards for the ethics of experimentation and research integrity. This is woefully under-reported in this paper.

Statistics: This is an insufficient rebuttal to the original question. First, you have a split plot design, correct? You have a factorial design to your dietary treatments, oils with and without betaine.

While I agree that PROC GLM remains an excellent and comprehensive procedure for working with fixed-effect-only linear models, PROC GLM is a fixed-effect only program, and PROC MIXED and subsequently PROC GLIMMIX, were developed to address known inadequacies of PROC GLM. Two cases where these inadequacies are especially evident are experiments with split plot features and experiments with repeated measures. PROC GLM was not designed to compute appropriate standard errors, test statistics, or confidence intervals for most mean comparisons of interest with split plot designs, and either cannot do so, or can only do so with programming heroics that are unnecessary given the ease with which PROC GLIMMIX or PROC MIXED handle the same analyses.

     Additionally, if you are missing any datapoints at all (which you did not address in the materials and methods or in the results), using GLM and treatment block and/or cat as fixed, your standard errors will be incorrect. The use of cat as a fixed effect presents an additional issue in that it limits the scope of inference to the cats used in the study, such that their findings can’t be generalized. As such, using a mixed model would allow you to identify cat as a random effect.

     Also, did you not use a FDR, like Turkey Kramer for example, for your non metabolomics data?

Author Response

Response to reviewer number 2

While the authors have added a statement regarding hypotheses, this is not a pointed hypothesis, this is very broad. Is there any reason why authors did not provide a hypotheses among treatments, specifically how n3 from fish vs. flax and with and without betaine? A hypothesis would be greater or less than or increased and decreased, stating that they will be different is not.

I do understand the reviewers point and have changed the article to highlight the alternate hypothesis that the reviewer suggests.  I think this will help the reader understand the background and the thinking in preparation for the test.  However, I’ve left in the hypothesis on which the statistics are based now stated as a null hypothesis as well as the alternate “greater or less than” hypothesis desired.  The null hypothesis is necessary as the alternate hypothesis is a one tailed hypothesis and we used and desired a two tail hypothesis as the desired statistical evaluation of the test. 

I understand that this is a study supported by Hills and likely a commercial formula, but stating that the foods had the same amount of “high concentration of high protein sources of chicken and corn gluten meal does not speak to the use of ingredients as the baseline. This is NOT sufficient and at minimum the ingredient deck of each formula needs to be added as a footnote. I agree that animals require nutrients rather than ingredients, but ingredients bring in different nutrients and the reader should know the relative compositions of these foods for a nutrition article. These must be added to allow the reader the opportunity to understand your dietary and as such, experimental treatments.

This is a good suggestion (to include the ingredient formulation).  These foods are not commercial and although Hill’s does add both flax and fish oil these addition amounts reflect only our desire for a specific fatty acid concentration and not any commercial product.  To the best of my knowledge the added amounts actually don’t reflect any commercial product concentration.    As requested, I’ve added the formulation which will allow the reader a better understanding of the foods.

Line 128: Choline chloride is horribly hydrophobic, did you ensure sufficient mixing of choline chloride?

Yes, this is standard industry practice.  Choline chloride is not only hydrophobic but is also quite caustic. 

Therefore, it is handled in a completely closed loop system from receipt to use. During food manufacture the 70% choline chloride aqueous liquid is metered directly into the pre-conditioner (which has a high water content) where it is mixed prior to extrusion. The food scientists who were in charge of manufacturing these foods have done validation studies in the past and also have continual monitoring of the system to ensure accurate delivery.

Carnitine, which you spelled incorrectly in your revised manuscript, is not available as L-carnitine for use in extruded food, usually it is added as carnitine tartrate, but the form is important to add. Further to this, the supplier of both the choline and carnitine needs to be stated.

Actually that is not a true statement or at least it wasn’t when the study was completed.   L-carnitine was available and was supplied by Lonza.  It was extruded and has 95%+ recovery which has been shown multiple times.  I suggest you taste the two for understanding of why l-Carnitine tartrate is not used here at these levels.  We’ve added the suppliers.

Line 139: This is problematic that not all cats were sedated and you do not disclose whether you used that in your statistical model. Sedation and stress alter the metabolize, your outcome measure. Also, describing the sedation is important as well. Please address this. Sedation affects intermediary metabolism and that is what you are measuring. In the future, I would recommend sedating all cats as this would reduce the variability among cats and addresses the animal welfare aspects and the human health and safety while working with this animal. At minimum you would need to discuss this lack of experimental control of a significant variable as a major limitation to the study.

Thank you for this line of questioning.  I was interested after your question in the first review.  In looking at what the sedation choices were and, as I suppose I should have expected, found the not surprising results that the technicians chose to sedate all of the cats before blood draw.  I’ve added the sedation information for clarity.

Housing and socialization was not addressed. The original comments was: “Line 122-127: This should be moved up in your materials and methods prior to the description of the conduct of the study. More information is required regarding housing. Were all cats housed in a single room? Provide details as to the room size, number of cats etc. If multiple rooms were used, did you consider this in your statistics? Furthermore, how did you feed treatment diets? Group housed cats are a challenge and it is unclear how these cats are individually fed, how intake was monitored etc.?”

     Your response is insufficient. You need to clearly define the housing and socialization that the cats received.  We know the cats are group housed, but nothing about how they were cared for and managed. This is important as  a role of the reviewer is to ensure that the research meets all applicable standards for the ethics of experimentation and research integrity. This is woefully under-reported in this paper.

We’ve added more information about the housing and care. 

Yes, multiple rooms were used.  No, housing is not considered in the statistical model as each treatment (n=8) was housed in one room with natural lighting and 24 hr access to an enclosed, glassed in porch (The main room measures 14'x14' and the porch measures 10'x7').  The cats have constant interaction opportunity with other cats, daily interaction with the technicians, multiple levels, and multiple toys.  It is true that should there be an effect of room this is confounded with the effect of treatment.  However, it is clear that one could not assign the increased betaine and single carbon metabolites in the betaine supplemented group, the increased apha linolenic concentration in the flax group or the increased circulating EPA/DHA in the fish oil group to location.  The rooms have the same foot print, the same lighting, the same porches.   I maintain it is appropriate to continue to use the individual cat as the experimental unit.  The cats are assigned to rooms in this fashion because of the very feeding system limitations mentioned.  Although a good approximation of food intake is measured in the feeding stations and access to the station is not allowed except as the computer reads the cat’s chip and opens the door this particular station had the possibility that a cat could scoop some food out to eat outside the station which also then made it possible for another cat to consume it.  As this would invalidate the one cat one food standard for this experiment we chose to use this housing assignment.  I’ve added the increased explanations as desired. 

Statistics: This is an insufficient rebuttal to the original question. First, you have a split plot design, correct? You have a factorial design to your dietary treatments, oils with and without betaine.

While I agree that PROC GLM remains an excellent and comprehensive procedure for working with fixed-effect-only linear models, PROC GLM is a fixed-effect only program, and PROC MIXED and subsequently PROC GLIMMIX, were developed to address known inadequacies of PROC GLM. Two cases where these inadequacies are especially evident are experiments with split plot features and experiments with repeated measures. PROC GLM was not designed to compute appropriate standard errors, test statistics, or confidence intervals for most mean comparisons of interest with split plot designs, and either cannot do so, or can only do so with programming heroics that are unnecessary given the ease with which PROC GLIMMIX or PROC MIXED handle the same analyses.

     Additionally, if you are missing any datapoints at all (which you did not address in the materials and methods or in the results), using GLM and treatment block and/or cat as fixed, your standard errors will be incorrect. The use of cat as a fixed effect presents an additional issue in that it limits the scope of inference to the cats used in the study, such that their findings can’t be generalized. As such, using a mixed model would allow you to identify cat as a random effect.

     Also, did you not use a FDR, like Turkey Kramer for example, for your non metabolomics data?

This is a good point.  That being that this is a split plot and gender cannot be assigned so it’s not a true block.  Also the standard errors reported although they represent the model defined do not actually represent the variance associated with the correct mean separation tests.  Therefore, the model in the original paper and the standard errors are incorrect (and does not account for missing data as was correctly pointed out).  Therefore, I have reanalyzed the data using the PROC MIXED procedure to better account for the split-plot design and correctly report the individual means standard errors.  We did use the Tukey-Kramer adjustment for multiple comparisons and have added that to our stats description.  I have reworked the tables to reflect this new analysis.

Round 3

Reviewer 1 Report

All my comments and concerned have been addressed. Thank you, I think the manuscript is very strong now.